# Benchmarking of shotgun sequencing depth reveals the potential and limitations of shallow metagenomics and strain-level analysis

Nicole S. Treichel [1] ✉, Charlie Pauvert [1], Joana Séneca [2,3], Petra Pjevac [2,3], David Berry [2,3], John Penders [4,5], Thomas C. A. Hitch [1,6] & Thomas Clavel [1,5] ✉

Shotgun metagenomics can provide both taxonomic and functional insights, but benchmarking is necessary to determine the sequencing depth appropriate for specific analyses. Here we used complex mixtures of DNA from cultured bacteria and analysed taxonomic composition, strain-level resolution and functional profiles at up to 11 sequencing depths (0.1–50.0 Gb). Reference-based analysis provided accurate strain-level taxonomy at 0.5–1.0 Gb. By contrast, de novo metagenome-assembled genome (MAG) reconstruction required deep sequencing (>10 Gb), and even MAGs deemed high quality by standard metrics were chimeric, with 54.5–81.8% accurately representing original strains, depending on the bioinformatic approach. Functionally, 2 Gb provided reliable insights at the pathway level for each of the mock communities tested, but sufficient proteome coverage was achieved only at or above 10 Gb. Library preparation and host DNA contamination were identified as confounders in shallow metagenomic analysis. This analysis highlights the potential and limitations of shallow metagenomics and provides guidance to accurately capture strain-level diversity using MAGs.

Next-generation sequencing is essential for microbiome research, as it enables comprehensive analysis of large datasets. Therefore, the robustness and accuracy of bioinformatic workflows for sequence analysis must be thoroughly assessed.

16S rRNA gene amplicon sequencing is widely used because it is cost-effective and easy to implement, but it provides taxonomic resolution typically limited to the genus level and is prone to amplification bias[1]. Shotgun metagenomics offers higher resolution and functional information, although it is more expensive and challenging, especially for low-biomass or host-DNA-rich samples.

Shallow metagenomics, commonly referred to as shotgun sequencing ≤1 Gb (~3 million reads), has been proposed to overcome the limitations of both 16S rRNA gene amplicon and deep metagenomic sequencing[2–4]. Few studies have benchmarked shallow metagenomics to determine what information can be reliably obtained at specific sequencing depths. Most compare techniques and depths using native samples, such as human stool, but their unknown composition limits accurate benchmarking[3–7]. Even fewer use defined mixtures of microorganisms or their DNA, hereon referred to as mock communities[2,8]. Other approaches include bioinformatic subsampling of deeply sequenced

[1]Functional Microbiome Research Group, Institute of Medical Microbiology, University Hospital of RWTH Aachen, Aachen, Germany. [2]Joint Microbiome Facility of the Medical University of Vienna and the University of Vienna, Vienna, Austria. [3]Division of Microbial Ecology, Department of Microbiology and Ecosystem Science, Centre for Microbiology and Environmental Systems Science, University of Vienna, Vienna, Austria. [4]Department of Medical Microbiology, Infectious Diseases and Infection Prevention, NUTRIM Institute for Nutrition and Translational Research in Metabolism, Maastricht University Medical Centre+, Maastricht, the Netherlands. [5]Euregional Microbiome Center, Maastricht, the Netherlands. [6]Department of Anaesthesia and Intensive Care, The Chinese University of Hong Kong, Hong Kong SAR, China. ✉e-mail: ntreichel@ukaachen.de; tclavel@ukaachen.de

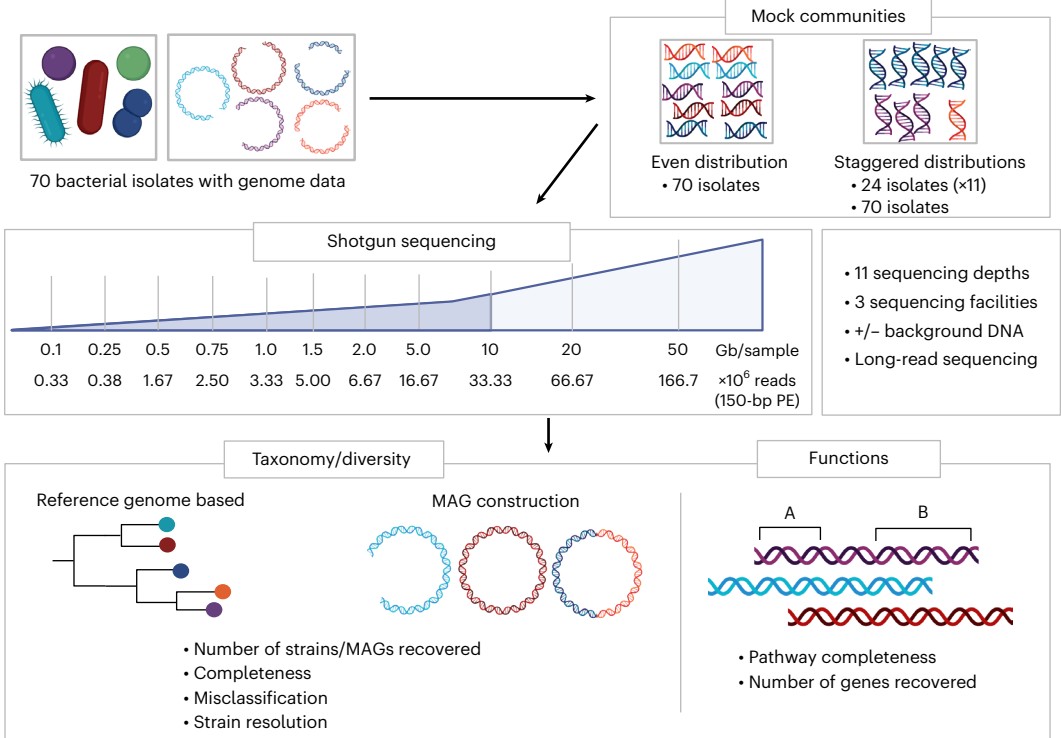

**Fig. 1 | Schematic overview of the experimental design.** The genomic DNA of 70 bacterial isolates (www.dsmz.de/miBC; https://www.hibc.rwth-aachen.de) was used as input to create 13 different mock communities. Two mock communities contained all members, one each with an even distribution (Mock-even-70) or staggered distribution (Mock-stag-70). In addition, mock communities with varying staggered distribution of 24 isolates were created: Mock-stag-24, sequenced at nine different sequencing depths; Mock-stag-24 v1-10 sequenced at 10 Gb for testing multicoverage binning for MAG construction. A second version of Mock-even-70 and one Mock-stag-24 were created by spiking DNA isolated from the gut content of germ-free mice. Libraries for these four mock communities (Mock-even-70 and Mock-stag-24 ± background DNA (bgDNA)) were prepared in two different sequencing facilities. A library per sequencing depth was then sequenced using the Illumina technology (short reads) at up to 11 sequencing depths. Mock-stag-70 was also sequenced using Oxford Nanopore Sequencing (long reads) in a third facility. Bioinformatic analyses included: (1) the number and relative abundance of strains; (2) the coverage of reference genomes; (3) the number and diversity of predicted proteins and completeness of functional pathways; and (4) the strain-level resolution using both a reference-based and MAG approach. PE, paired ends. Figure created in BioRender; Treichel, N. https://biorender.com/k34g802 (2026).

datasets or the creation of in silico mock communities; however, these methods do not account for biases introduced during laboratory processing and sequencing[3,6,9].

To better define the strengths and limitations of shallow metagenomics, and provide guidance for future studies, we systematically evaluated the effects of shotgun sequencing depth on composition, strain-level diversity and functional readouts (Fig. 1). We used 13 complex mixtures of bacterial DNA: an even distribution with 70 bacterial strains (Mock-even-70), a staggered distribution with 24 strains (Mock-stag-24), including 10 additional versions with different compositions (Mock-stag-24 v1-10), and a staggered distribution with 70 strains (Mock-stag-70) (Extended Data Fig. 1 and Supplementary Data 1). To account for wet-lab and host factors, DNA libraries were prepared in two laboratories and in the presence/absence of background DNA isolated from the gut of germ-free mice.

## Results

### Shallow metagenomics enables reliable reference-based taxonomic profiling

We first investigated the impact of sequencing depth on taxonomic composition. For the three main mock communities (Mock-even-70, Mock-stag-24 and Mock-stag-70), reads were detected for all reference genomes already at 0.1 Gb. To obtain an overview of genome coverage, completeness categories were compared between the sequencing depths. At 0.1 Gb, most of the genomes (63–91%) had a low coverage (0–25%) (Fig. 2a). Genome coverage increased substantially until 5 Gb of sequencing, with a clear effect of mock community complexity. Most genomes reached >90% coverage at 5 Gb in Mock-even-70

and Mock-stag-24. However, in Mock-stag-70, coverage continued to increase gradually from 5 Gb (36% of genomes at >90% coverage) to 50 Gb (64%), although 11 genomes—added at 0.001 ng (0.00046‰) to 0.1 ng (0.0046‰)—still had low coverage (0–25%) at the latter sequencing depth.

Looking at individual strains (Extended Data Fig. 3), the lowest coverage in Mock-even-70 at 0.1 Gb was 3.34% for *Phocaeicola sartorii* CLA-AV-12 and the highest 58.7% for *Bacteroides uniformis* CLA-AV-11, which reached 99% coverage already at 0.75 Gb. Full genome coverage (100%) was achieved only at 10 Gb for *Veillonella intestinalis* CLA-AV-13. In Mock-stag-24 and Mock-stag-70, higher coverage was associated with increasing DNA concentration. The reference genomes with the lowest concentration in Mock-stag-24 (0.01%, 0.04 ng) had a minimum coverage of 0.53% at 0.1 Gb and a maximum coverage of 34.21% at 10 Gb. In Mock-stag-70, the low-abundance genomes—0.1 ng (0.0046‰) to 0.001 ng (0.00046‰)—reached 0.8–21.8% coverage at the highest sequencing depth (50 Gb), whereas the genomes of the five strains with the highest DNA amounts (>4.6%) achieved 40.50–98.25% coverage already at 0.1 Gb.

Next, we assessed the relative abundance of strains and compared them with their theoretical values. The overall relative abundance profiles of Mock-even-70, Mock-stag-24 and Mock-stag-70 showed no significant impact of sequencing depth ($P$ = 0.78, 1.0 and 1.0, respectively; Kruskall–Wallis test) (Supplementary Figs. 1 and 2). In addition, bioinformatic subsampling of Mock-stag-70 sequenced at 50 Gb (ten replicates) showed that variations in the relative abundance of individual species were low (average coefficients of variation <5%), with increasing variability at lower sequencing depths (Fig. 2b; see statistics in Supplementary Data 5).

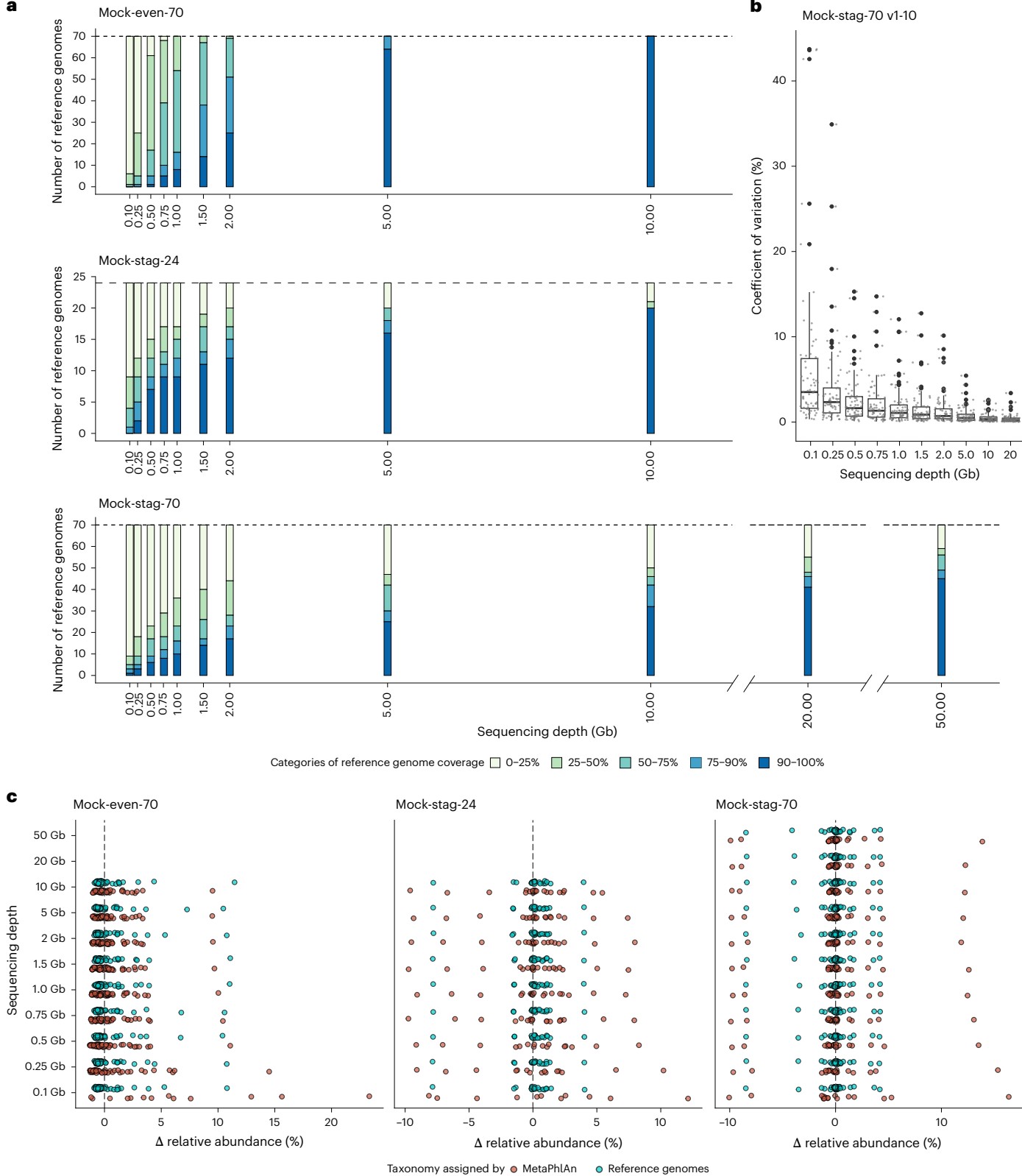

**Fig. 2 | Reference-based taxonomic profiles. a**, Number of reference genomes per category of coverage by metagenomic reads (colour gradients) for Mock-even-70, Mock-stag-24 and Mock-stag-70 (top to bottom) at up to 11 sequencing depths (x axis). **b**, Coefficients of variation of relative abundances of the 70 strains in 10 in silico datasets for each sequencing depth, subsampled from Mock-stag-70 sequenced at 50 Gb. The coefficients of variation between the sequencing depths were tested statistically using a Kruskal–Wallis rank-sum test with Benjamini–Hochberg correction (see P values in Supplementary Data 5). The lower and upper border of the boxes represent the 25th and 75th percentile,

and the centre line indicates the median. Whiskers represent the highest and lowest values excluding outliers. Large black points represent outliers. Small grey points show the coefficients of variation between the ten mock communities for the 70 strains at each sequencing depth. **c**, Difference (delta values) between measured relative abundances and theoretical values (x axis) for the three mock communities after taxonomic assignment using either the reference genomes (blue) or MetaPhlAn4 (red). In the MetaPhlAn analysis, only the taxa that matched reference genomes at the species level were considered.

As reference genomes are not available for metagenomic analysis in most studies, the mock communities were also taxonomically analysed using the commonly used profiler MetaPhlAn[10], hereon referred to as the non-supervised approach. Overall, the taxonomic assignment in this approach was less sensitive, that is, fewer species were detected. In Mock-stag-24, one strain with low concentration (0.1%, 0.4 ng, *Hominilimicola fabiformis* CLA-AA-H232) was not detected (Supplementary Fig. 4). In Mock-even-70, 59 species were assigned using MetaPhlAn4, including 47 that matched a reference genome at the species level. In Mock-stag-70, 53 taxa were detected, 44 of which had a species-level match (Supplementary Figs. 3 and 5). The non-supervised approach also showed higher variance from the targeted relative abundances, particularly at lower sequencing depths and with increasing complexity of the mock communities (Fig. 2c; see statistics in Supplementary Data 5).

In summary, reference-based detection of strains is possible with as little as 0.5 Gb of sequencing, but achieving high coverage of reference genomes requires more data (>5 Gb) under the conditions tested. Relative abundance profiles were not substantially affected by sequencing depths for most of the strains. By contrast, non-supervised taxonomic assignment was less sensitive, with fewer species detected.

## De novo strain level analysis requires deep sequencing and generates chimeras

Strain-level resolution was evaluated using four *Escherichia coli* and *Phocaeicola vulgatus* strains in Mock-even-70 (Extended Data Fig. 1). All strains were detected at 0.1 Gb and achieved >75% genome coverage at 5 Gb and >98% at 10 Gb (Extended Data Fig. 4a). *P. vulgatus* strains showed similar coverage increase, reflecting their high average nucleotide identity (ANI) values (Extended Data Fig. 4b). For *E. coli*, the two strains with highest ANI (99.97%) displayed overlapping coverage curves, whereas the more divergent strain CLA-AD-1 (ANI <97%) exceeded 99% coverage already at 2 Gb. By contrast, MetaPhlAn4 detected only one strain per species across all sequencing depths in both Mock-even-70 and Mock-stag-70 (Supplementary Figs. 3 and 5).

Because reference datasets are often unavailable, genomes were reconstructed de novo by assembling metagenome-assembled genomes (MAGs) in each sample. The number of MAGs increased with sequencing depth across all mock communities (Fig. 3a and Extended Data Fig. 5). At high depths, more MAGs than reference genomes (dashed line) were recovered, yet some references remained unrepresented. The number of MAGs per reference genome also rose with sequencing depth, with ≥2 MAGs observed for 22, 12 and 29 references in Mock-even-70, Mock-stag-24 and Mock-stag-70, respectively (Extended Data Fig. 6), indicating splitting into multiple MAGs rather than coalescence into single high-quality genomes.

To clarify the unexpected number of MAGs and their multiplicity per genome, we assessed their origin by grouping them by quality and assigning contigs to reference genomes (>0.25% coverage threshold). In Mock-even-70 at 10 Gb, 10 of 23 high-quality MAGs (hqMAGs; completeness >90%, contamination <5%) were assigned to

multiple genomes (Fig. 3b), with multigenome assignments increasing as MAG quality decreased. GTDB taxonomy agreed with assignments to the reference genomes (Supplementary Data 4). Despite four *E. coli* strains in Mock-even-70, only one MAG (assigned to *E. coli* CLA-AD-1) was recovered per dataset >1 Gb (except at 2 Gb, where an additional low-quality MAG appeared). Similarly, although four *P. vulgatus* strains were present, only one MAG was reconstructed at 5 and 10 Gb, primarily assigned to *P. vulgatus* strain HDF; therefore, strain delineation was not achieved (Supplementary Data 4).

In Mock-stag-24, a single hqMAG was assembled at 0.75 and 1 Gb (Fig. 3b), primarily assigned to the genome with the highest concentration (40 ng, *Thomasclavelia ramosa* CLA-JM-H52; ≥95% coverage). With increasing sequencing depth, hqMAGs from lower abundant genomes emerged (Supplementary Data 4). Unexpectedly, single-origin MAGs were more frequent at lower quality, whereas at 10 Gb, half of the hqMAGs were chimeric.

To illustrate chimerism, three hqMAGs (Mock-even-70, 10 Gb) were aligned to the reference genomes they covered >0.25% using blastn (Fig. 3c). MAG 54 matched a single genome (>98% coverage), MAG 16 included fragments of a second genome, and MAG 40 was highly chimeric, comprising sequences from 12 genomes (10 of which are shown, ranked by decreasing coverage from the outer to the inner circle). At 10 Gb, the mean/maximum number of reference genomes per hqMAG was 2.3/12 (Mock-even-70), 1.5/2 (Mock-stag-24) and 2.1/5 (Mock-stag-70) (Supplementary Data 4).

As multicoverage binning was shown to increase the number and quality of MAGs[11], we sequenced (10 Gb) ten additional Mock-stag-24 communities (v1-10) with varying reference genome distribution (Supplementary Data 1). Compared with single-coverage binning, significantly more hqMAGs per mock community were assigned to only one reference genome using multicoverage binning (Fig. 3d, blue dots), but chimeric MAGs still occurred in half of the communities (Fig. 3d, orange dots).

To further assess ways to enhance MAG reconstruction, the most complex community (Mock-stag-70) was used to compare assemblers (MEGAHIT versus metaSPAdes) and sequencing technologies (Illumina short reads versus Nanopore long reads). MetaSPAdes assemblies produced more single-origin hqMAGs, especially ≥10 Gb (+1 at 10 Gb, +10 at 20 Gb, +5 at 50 Gb), a result confirmed by in silico subsampling analyses (Fig. 3e,f). Long-read sequencing yielded the highest proportion of single-reference hqMAGs (14 of 15 at 10 Gb; 18 of 22 at 50 Gb) (Fig. 3g).

To test whether chimeric sequences are generated already during the assembly process, contigs were assembled from the 10, 20 and 50 Gb data using either MEGAHIT or metaSPAdes and were aligned to the reference genomes using blastn (Extended Data Fig. 7). Most contigs mapped to a single reference species (average of three sequencing depths: MEGAHIT, 91.1 ± 1.6%; metaSPAdes, 93.5 ± 1.0%), suggesting that 6.5–8.9% misassembled contigs contributed to the reconstruction of chimeric MAGs.

In summary, when high-quality references are available, read mapping enables strain-level analysis of shallow metagenomes. By contrast, de novo MAG reconstruction produces chimeric genomes—even at supposed high quality—due to both assembly and

**Fig. 3 | Strain analysis. a**, Strain analysis based on the MAGs assembled in Mock-even-70 (top) and Mock-stag-24 (bottom) at each sequencing depth individually. The number of bins assembled from the shotgun data is shown (left, dark blue), as well as the number of reference genomes matching MAGs (coverM) (right, lighter blue), indicating that multiple MAGs were reconstructed for some mock species. The reference genome with the highest coverage by MAG reads was chosen as representative of that MAG. **b**, Number of high-, medium- and low-quality MAGs assigned to either one (blue) or several (red) reference genomes (with coverM using a cut-off of >0.25% coverage of reference genomes). **c**, Three exemplary high-quality (>90% completeness, <5% contamination) (hq) MAGs (dark grey, outer circle) reconstructed from Mock-even-70 at 10 Gb. They were aligned to the reference genomes they covered by more than 0.25% to illustrate different categories of chimerism. The predominantly covered reference genome is depicted in blue, while chimeric sequences of further reference genomes

(inner circles) are shown in red. **d**, Number of hqMAGs assigned to one reference genome (blue) or more reference genomes (orange) for the ten different Mock-stag-24 (v1-10) with varying reference genome abundance distribution. MAGs were constructed with either a single-coverage (bright colours) or a multicoverage (darker colours) binning approach. Statistics: Kruskal–Wallis rank-sum test; ***$P$ = 0.0003871. NS, not significant. **e**, Number of hqMAGs constructed from contigs assembled with either MEGAHIT or metaSPAdes for Mock-stag-70. The MAGs were then assigned to one (green) or more (red) reference genomes as above. **f**, High-quality MAGs binned from assemblies generated with either MEGAHIT or metaSPAdes using ten datasets at 10 Gb subsampled in silico from 50 Gb. Bars are mean values; whiskers are standard deviations. Statistics: Wilcoxon rank-sum test (two-sided); *$P$ = 0.012. **g**, Number of hqMAGs binned from assemblies acquired by long-read sequencing of Mock-stag-70 and categorized as in **e**.

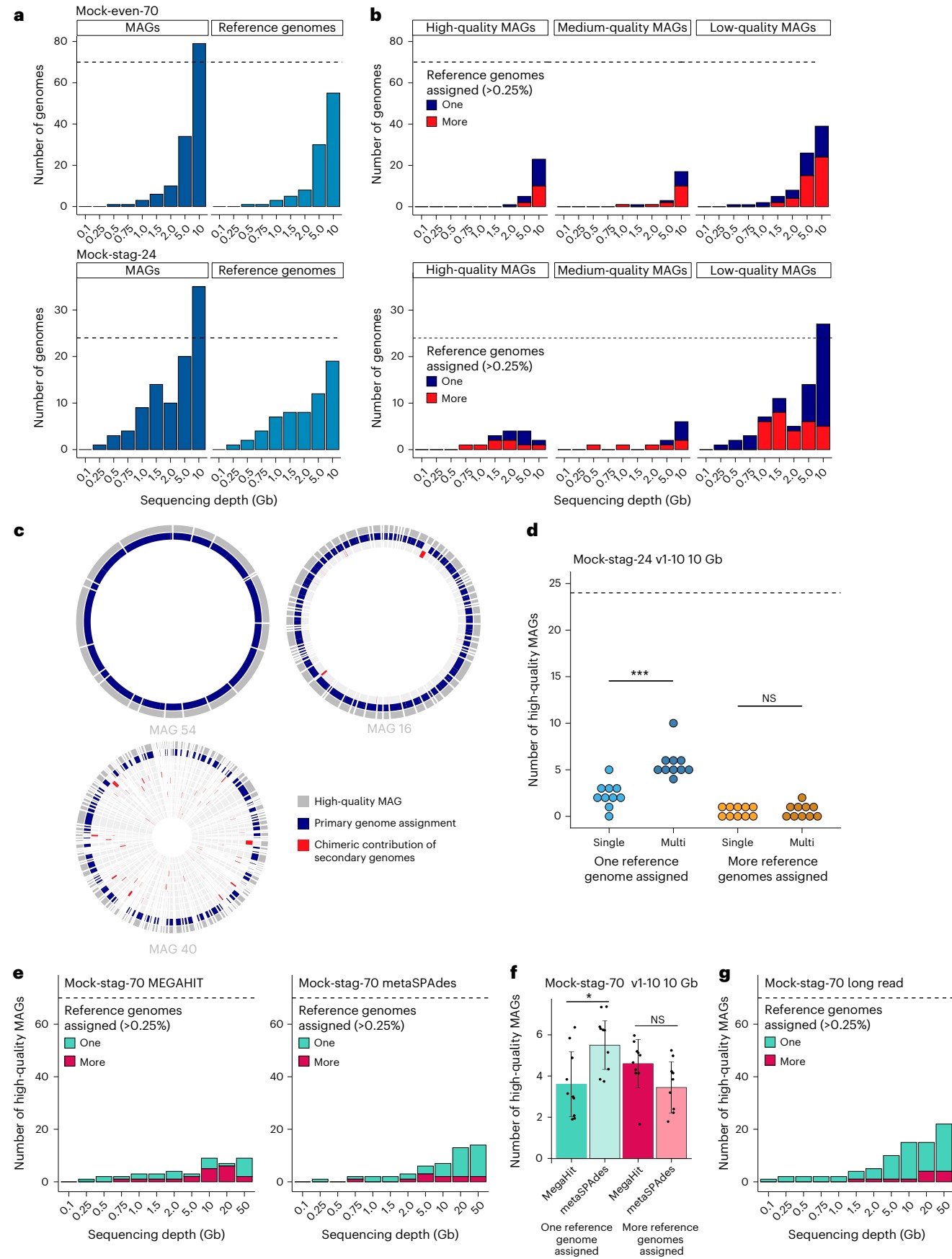

binning. Chimerism was partially reduced by multicoverage binning, strain-aware assembly (metaSPAdes) and long-read sequencing. These findings suggest that strain diversity in MAG catalogues is artificially inflated.

### Functional coverage is limited by shallow sequencing

Because functional profiling motivates metagenomics over 16S rRNA gene amplicon sequencing, we evaluated sequencing depth requirements for pathway and gene-based analyses.

Kyoto Encyclopedia of Genes and Genomes (KEGG) pathway completeness (percentage of detected KEGG Orthologs per pathway) was assessed for pathways present in the reference genomes (Extended Data Fig. 8). Among 178 pathways included in the analysis by KEGG-Decoder[12], 121 (Mock-even-70, Mock-stag-70) and 118 (Mock-stag-24) were represented in the references. At 10 Gb, a maximum of 80 (Mock-even-70), 77 (Mock-stag-24) and 81 (Mock-stag-70) pathways were complete with the two assembly methods tested (Extended Data Fig. 8). One pathway was not detected in staggered mocks, and 39–41 pathways remained incomplete. Average pathway completeness increased with sequencing depth (Fig. 4a), exceeding 80% at 2 Gb (Mock-stag-70) and 5 Gb (others), then plateauing (+3.7–7.8% at maximum depth).

To account for functionally unassigned proteins, detected protein sequences were compared with the repertoire of the reference genomes (Mock-70: 227,543; Mock-stag-24: 82,992 protein sequences). Protein sequence recovery rose to 94.7% in Mock-even-70 (+22.9% from 5 to 10 Gb) but plateaued at 5 Gb in Mock-stag-24 (+12.9%) (Fig. 4b). In Mock-stag-70, coverage reached 55.5/58.3% at 10 Gb and increased only to 71.8/73.8% at 50 Gb (MEGAHIT/metaSPAdes). In silico subsampling of Mock-stag-70 (0.1–20 Gb) confirmed depth-dependent functional recovery with low replicate variation (Fig. 4c; Wilcoxon rank-sum test: Supplementary Data 5).

In summary, ~5 Gb sufficed for functional pathway-level analyses across mock communities, whereas comprehensive protein-level recovery required greater sequencing depth depending on community complexity.

### Impact of sample processing and background DNA

To assess wet-lab effects on shallow metagenomics results, DNA libraries were prepared in two facilities using different protocols. In addition, Mock-even-70 and Mock-stag-24 were spiked with DNA isolated from gut content of germ-free mice to simulate host DNA, or left unspiked.

Relative abundance profiles showed distinct clustering owing to both factors (background DNA, facility), with more pronounced effects linked to differences in library preparation protocols between facilities (Fig. 5a). Within a condition (background DNA/facility pair), the extreme sequencing depths (0.1 and 10 Gb) tended to be most distant from each other. The relative abundance profiles of Mock-stag-24 prepared in facility 1, which used more template DNA (100 ng versus 1 ng) and a lower number of polymerase chain reaction (PCR) cycles (5 versus 12), were less sensitive to the effects of sequencing depth when background DNA was present (Fig. 5a, bottom).

Wet-lab effects on reference genome coverage were assessed using delta values between facilities and background DNA conditions. For Mock-even-70, profiles were consistent at 5–10 Gb regardless of facility or background DNA (Fig. 5b), whereas at ≤2 Gb, coverage varied by up to 39% between facilities and 13% with background DNA. For Mock-stag-24, interfacility variation was additionally influenced by input DNA quantity (Fig. 5c). With a reference genome input >1 ng and >1 Gb depth, variance between facilities dropped below 3.69%. Background DNA showed a similar effect on reference genome coverage, but with a more pronounced influence of the amount of reference genome input DNA: variance was negligible (1.33%) with ≥10 ng DNA and ≥1.5 Gb sequencing depth. For >1 ng DNA per strain, ≥5 Gb was required (total variance 3.23%).

Strain-level relative abundances in Mock-even-70 were mostly consistent between library preparation strategies (Supplementary Fig. 1a), indicating that differences shown in Fig. 5a stem from changes in the relative abundance of a few strains. In Mock-stag-24, deviations from theoretical abundances differed between facilities (Supplementary Fig. 1b): *Enterocloster clostridioformis* matched expectations in facility 1 but was most abundant in facility 2, whereas *T. ramosa* dominated in facility 1. This suggests that library preparation, particularly template DNA input and PCR cycles, affected relative abundance profiles.

We further assessed wet-lab effects on predicted protein profiles. Facility 1 yielded more complete functional profiles for both mock communities (Extended Data Fig. 9). Background DNA reduced coverage, although this effect decreased at high sequencing depths in facility 1. Overall, background DNA had a stronger impact with the facility 2 protocol.

In summary, using higher template DNA input and fewer PCR cycles improved the robustness of taxonomic and functional profiles. Increased sequencing depth can partially compensate low DNA input or the presence of non-target DNA.

## Discussion

In this work, we investigated the potential and limitations of shallow metagenomic sequencing using complex mock communities and sequencing depths from 0.1 Gb (0.3 million reads) to 50 Gb (166.7 million reads), with a specific focus on strain-level analysis.

Our results corroborate previous findings, that taxonomic profiles based on shallow shotgun sequencing (0.5–4 million reads) can reflect those obtained at a higher sequencing depths[2,3,6,8]. In our study, reads for all strains were detected already at 0.1 Gb (0.3 million reads) using a reference-based analysis, enabling prevalence and relative abundance analyses despite overall low genome coverage. This aligns with Hillmann et al. and Xu et al., who found that 0.15 and 0.5 Gb, respectively, was sufficient for taxonomic analysis of the human gut microbiome[2,3]. This makes shallow metagenomics at 0.5 Gb per sample attractive for cost-efficient diversity and composition profiling of large studies using a high-quality reference database.

Metagenomics enables high-resolution taxonomic analysis, surpassing the limitations of 16S rRNA gene amplicon sequencing. Xu et al. recovered 62% of 62 species at 1 Gb per sample using MetaPhlAn2 and the Refseq database[2]. Our analysis of mock communities with MetaPhlAn4 at 1 Gb showed similar results (Mock-stag-24, 75%; Mock-even-70, 67%; Mock-stag-70, 43%), without reaching strain-level resolution. Using reference genomes, we detected reads for all strains in mock communities at 0.1 Gb. Discrimination of strains with up to 99.66% ANI was possible, although sequencing depths >5 Gb are required for nearly full genome coverage, depending on strain abundance and genome size.

A sequencing depth ≥5 Gb was necessary to reconstruct sufficient MAGs, consistent with previous findings[3,6]. Importantly, at high sequencing depth (10–50 Gb), more MAGs were created than input strains, yet not all strains were represented. A substantial number of hqMAGs were chimeric, containing fragments from multiple reference genomes, despite excluding contigs <1,500 bp during binning to reduce chimerism. These results echo previous reports that ~5% of genes within MAGs differ from the dominant reconstructed taxa[13]. A recent study, which evaluated the quality of short-read assemblies from a soil community by mapping to the respective long-read data, reported that assembly failures occurred in most genome bins[14], supporting our findings. Our results show that deeper sequencing does not guarantee more complete MAGs and does not eliminate chimerism. Data from the staggered mock communities indicate that having more reads for a reference genome does not guarantee that it will be represented by a MAG, nor that the MAG will be of high quality. Instead, we observed the creation of several MAGs for the same reference genome. Overabundance of reads leading to high-coverage misassemblies has been observed previously[14].

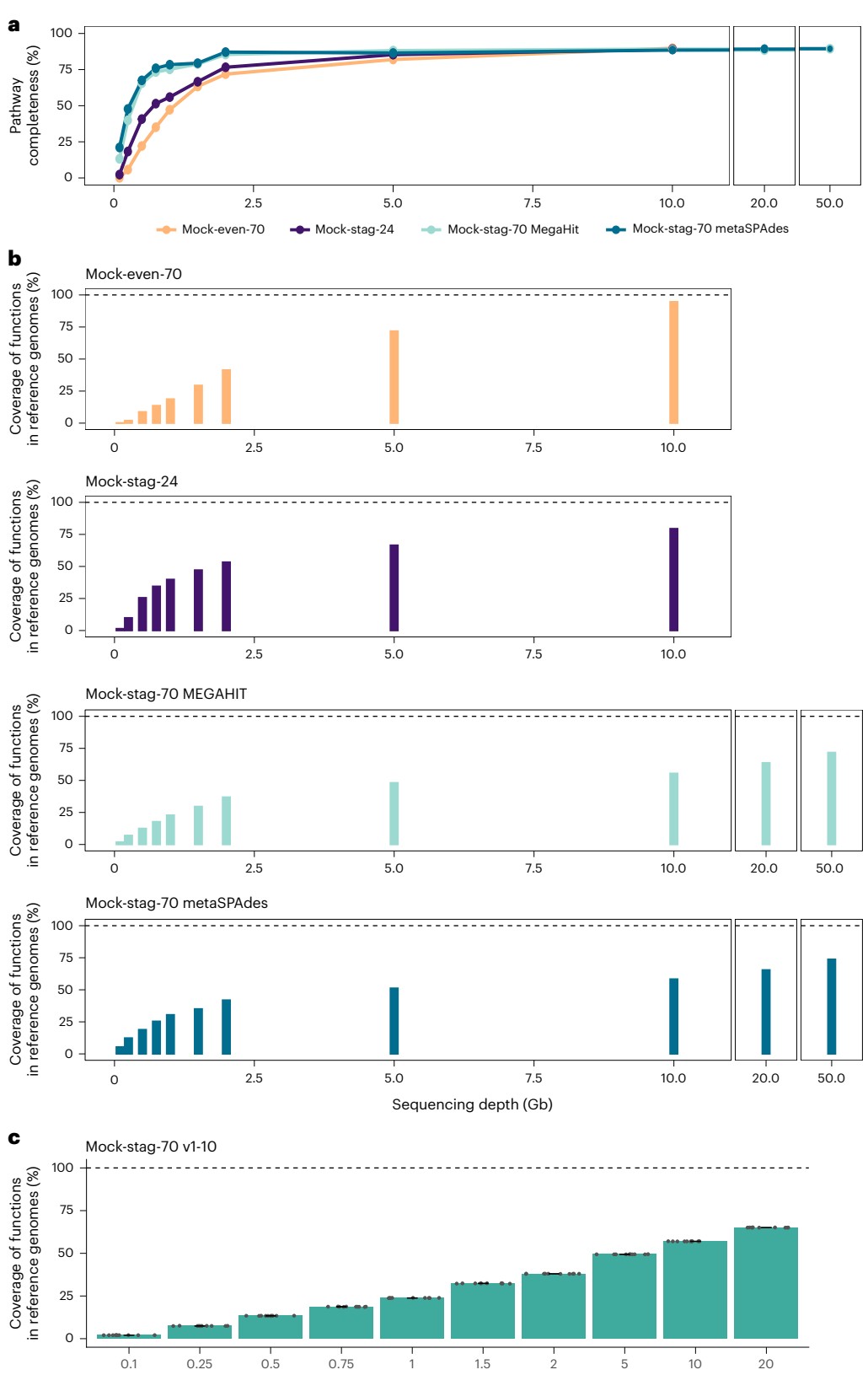

**Fig. 4 | Functional coverage. a**, Average KEGG pathway completeness of Mock-even-70, Mock-stag-24 and Mock-stag-70 using contigs assembled with MEGAHIT, and Mock-stag-70 using contigs assembled with metaSPAdes. **b**, Top: fraction of predicted proteins from the reference genomes covered by the metagenomic assemblies (MEGAHIT) for three different mock communities at the respective sequencing depths. Bottom: reads of Mock-stag-70 were additionally assembled using metaSPAdes. **c**, Average coverage of functions of the reference genomes by ten in silico datasets, subsampled to ten different sequencing depths each from Mock-stag-70 sequenced at 50 Gb. Bars are mean values; whiskers are standard deviations. Statistics: Kruskal–Wallis rank-sum test with Benjamini–Hochberg correction (see P values in Supplementary Data 5).

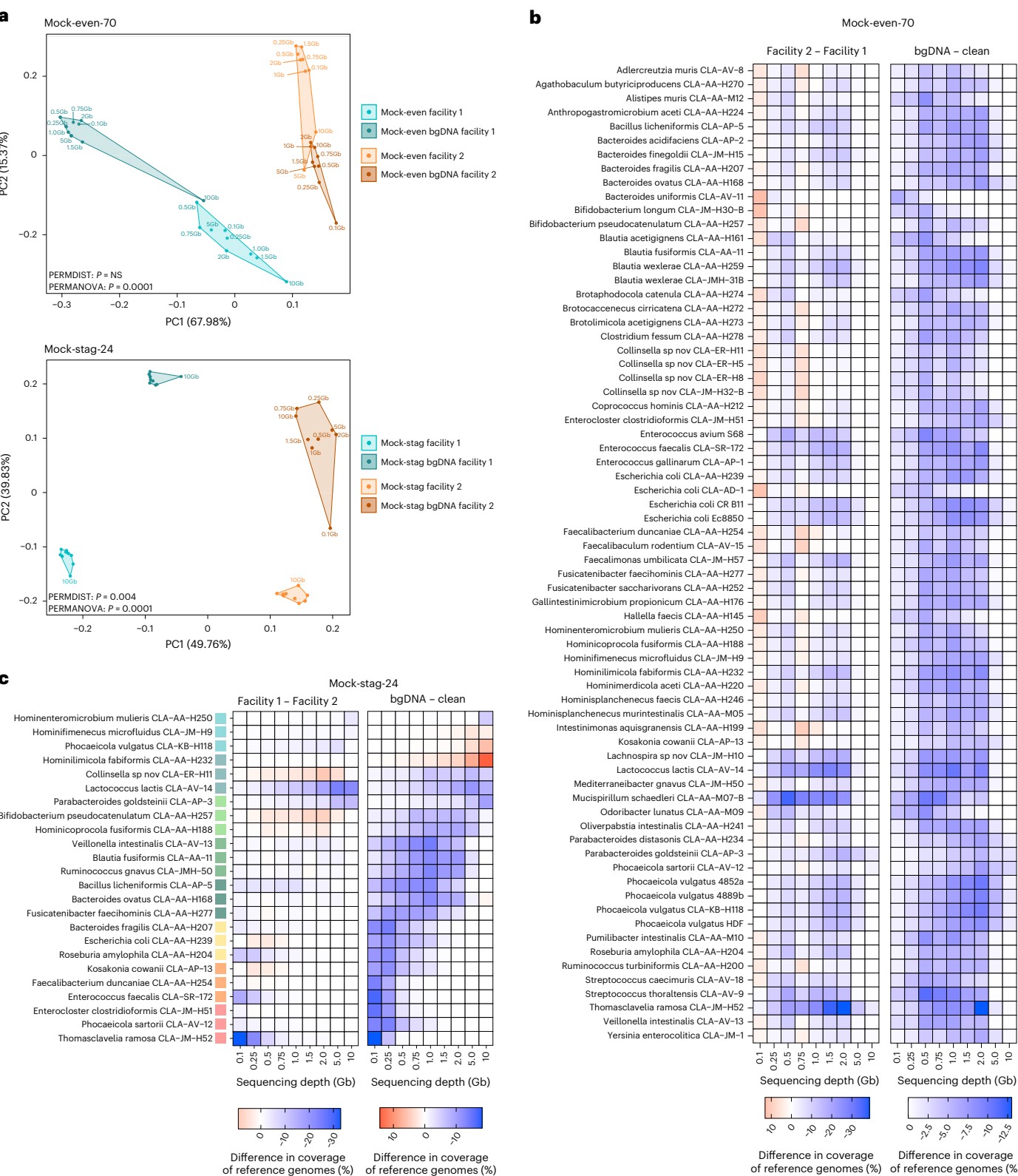

**Fig. 5 | Effects of background DNA (bgDNA) and library preparation in two facilities. a**, Principal component analysis (PCA) plot of relative abundance profiles in Mock-even-70 (top) and Mock-stag-24 (bottom). PERMDIST, analysis of multivariate homogeneity of group dispersions; PERMANOVA, permutational multivariate analysis of variance. **b**, Difference in coverage of reference genomes between facility 1 and facility 2 (left) and samples with or without bgDNA (right) for Mock-even-70. **c**, As in **b** for Mock-stag-24. The reference genomes were ranked (from top to bottom) according to increasing DNA amount in the mixture, as indicated by the colour gradient (from blue to red; 0.04, 0.4, 1, 2, 4, 10, 20 and 40 ng).

Bioinformatic approaches reducing chimeric hqMAGs in our work were multicoverage binning and strain-aware assembly, which is in line with previous studies[11,14]. Long-read sequencing also increased the number of coherent hqMAGs. This highlights the need to consider long-read or hybrid sequencing for projects where the goal is strain reconstruction, and to enhance such approaches further[15]. Nevertheless, none of the approaches resolved the issue completely, and the data presented raise concerns about the prevalence of spurious sequences in ever-growing MAG catalogues[16,17], as we used common bioinformatic workflows to create them.

To assess functional information loss with decreasing sequencing depth, we calculated KEGG pathway completeness, showing that >2 Gb is required for >50% completeness. By contrast, a study on a mock community of 62 human gut bacteria reported that 1 Gb of shallow metagenomics gave a functional profile similar to the one obtained from the reference genomes at KEGG levels 1 and 2 (ref. [2]). Another study based on in silico subsampling of an ultradeep metagenome (2.5 billion reads; 750 Gb) to 0.5 million reads (0.15 Gb) showed a nearly full recovery of KEGG Orthology groups[3]. This result might be influenced by different complexity of communities and their subsampling by repeated rarefying creating a higher chance of recovering functions. However, our data suggest that shallow metagenomics is not suitable at the level of predicted protein sequences, as <75% recovery occurred at depths ≤5 Gb, consistent with a study showing that at least 24 Gb was needed to recover the full richness of antimicrobial resistance gene families[18].

The influence of wet-lab factors has previously been compared for traditional metagenomic sequencing of native samples and mock communities, revealing that the community diversity, amount of input DNA, and the sequencing platform alter the results[19–21]. Regarding sequencing depth, library preparation and host background DNA impacted taxonomic and functional results at <5 Gb. Strains with higher abundance were less affected, even at lower sequencing depths. In case of a high amount of background DNA or variation of protocol, higher sequencing depth may improve the robustness of results.

This study has some limitations. (1) Different environments may require different metagenomics strategies. Here, we focused on wet-lab parameters related to gut microbiomes. Although showing their overall effect on metagenomic data, the study design was not appropriate to disentangle the effect of each parameter separately. Previous studies have investigated other aspects such as low biomass, contamination or additional library preparation protocols (DNA input, PCR cycles and fragment/insert size)[19,22]. (2) Although we used MetaPhlAn4 for some of the analyses, using reference genomes in most instances represents an ideal case for taxonomic and functional readouts. (3) Despite testing multiple common standard approaches for studying chimeric MAGs, we cannot rule out that additional, untested strategies may perform better. Future benchmarking of bioinformatic tools, as in Critical Assessment of Metagenome Interpretation (CAMI), is necessary to address chimeric MAGs at scales that go beyond our study[23]. (4) The mock communities included only bacteria, excluding the influence of other microorganisms. (5) We used a higher number and more complex mock communities than in previous studies; nonetheless, they are only approximations to the ~300–400 bacteria species native to the gut of an individual.

In summary, as few studies benchmark shallow metagenomics with known input, many users may not recognize its limitations. Sequencing depth should align with study goals and consider microbial diversity, evenness and available DNA. Shallow metagenomics can be suitable for large studies aiming at database-base guided taxonomic profiles of well-characterized environments, but it is unsuitable for high-resolution functional analysis or de novo strain resolution. Even with deep sequencing, MAG-based approaches require careful consideration to minimize the generation and spread of artificial bacterial diversity.

## Methods

### DNA extraction and preparation of the mock communities
For the creation of the mock communities (Mock-even-70, Mock-stag-24 and Mock-stag-24 v1-10 for multicoverage binning, and Mock-stag-70), DNA was obtained from isolates in our collections of mouse and human gut bacteria[24,25] (Extended Data Fig. 1). DNA was extracted from freshly grown strains revived from frozen glycerol stocks based on the method of Godon et al.[26] modified as described by Afrizal et al.[24]. DNA concentration was measured using a Qubit fluorometer (Thermo Fisher Scientist). The DNA of all 70 isolates was pooled in either equimolar amounts (Mock-even-70), or in a staggered distribution (Mock-stag-70). For 24 selected isolates, the DNA was pooled in a staggered distribution (Mock-stag-24). For testing multicoverage binning to generate MAGs, 10 additional mock communities with 24 DNA extracts in 10 different distribution patterns were created (Mock-stag-24 v1-10). The distribution of DNA from the different isolates in each mock community is provided in Supplementary Data 1.

Non-bacterial background DNA was isolated, as described above, using gut content collected from germ-free mice. After DNA extraction, the background (non-target) DNA was mixed 1:1 (v/v) with Mock-even-70 and Mock-stag-24.

### Mouse samples
Germ-free C57BL/6J mice were bred in the gnotobiotic unit of the Institute of Laboratory Animal Science at the University Hospital of RWTH Aachen under ethical approval (LANUV no. 81-02.04.2023.A253) and in accordance with the German Animal Protection Law (TierSchG). Room temperature was kept between 21 °C and 24 °C and 25–40% humidity on a 12 h:12 h day:night cycle. All mice were fed a standard chow ad libitum (γ-irradiated ssniff Spezialdiäten ref. V1124-927). The germ-free status of the mice was confirmed by microscopic observation after Gram staining and by cultivation on both anaerobic and aerobic agar plates. Mice were culled, and caecal content was collected and stored immediately at −80 °C before DNA extraction as described above.

### Library preparation and short-read sequencing
In sequencing facility 1 (UKA), DNA libraries of the mock communities were prepared using the NEBNext Ultra II FS DNA Library Prep Kit for Illumina (NEB) according to the manufacturer's instructions, using 100 ng (Mock-even-70 and Mock-stag-24), 200 ng (Mock-stag-70) or 40 ng (Mock-stag-24 v1-10 for multicoverage binning) of input DNA and an automated platform (Beckman Coulter). Enzymatic shearing to approximate 250 bp was performed for 30 min. Adaptor-ligated DNA was enriched using PCR (Mock-even-70, Mock-stag-24, Mock-stag-70: 5 cycles; Mock-stag-24- v1-10 multicoverage binning: 7 cycles) and NEBNext Multiplex Oligos for Illumina (NEB) for unique dual barcoding. AMPure beads (Beckman Coulter) were used for size selection and clean-up of adaptor-ligated DNA.

Sequencing facility 2 (UMC) used the Nextera XT DNA Library preparation kit (Illumina) with Nextera XT indexes (Illumina) and 1 ng template DNA according to manufacturer's protocol. Twelve cycles of PCR were used for indexing. AMPure beads (Beckman Coulter) were used for double-sided selection and clean-up of adaptor-ligated DNA.

For cleaned DNA library from both facilities fragment size (~320 bp library fragment size, ~200 bp genomic DNA insert size) was determined on an Agilent D1000 Tapestation (Bioanalyzer System, Agilent Technologies) using High Sensitivity D1000 screentapes.

Quality check (Bioanalyzer System, Agilent Technologies), DNA quantification (Quantus, Promega) and sequencing of the resulting libraries were conducted at the IZKF Core Facility Genomics (UKA, RWTH Aachen University). The libraries for Mock-even-70, Mock-stag-70 and Mock-stag-24 were pooled to reflect the 9 or 11 different sequencing depths targeted, and sequenced on a NovaSeq6000 (Illumina) with NovaSeq 6000 Reagents v1.5 (2 × 150 cycles). Mock-stag-24 v1-10 for multicoverage binning was sequenced at

10 Gb with the same chemistry at the NGS Competence Center Tübingen (NCCT).

## Long-read sequencing

The Mock-stag-70 DNA (200 fmol) was prepared for long-read sequencing with the SQK-LSK114 kit (Oxford Nanopore Technologies), following the manufacturer's protocol and the NEBNext Companion module v2 (New England Biomedicals). Approximately 80 fmol of a >20-kb library was loaded onto an R10.4.1 flowcell (FLO-PRO114, Oxford Nanopore Technologies) and sequenced for 72 h on a Promethion P2 solo (Oxford Nanopore Technologies) using MinKNOW v.25.05.12. The flowcell was washed using the EXP-WSH004 kit, and leftover library was reloaded and left to sequence until the flowcell end of life. Reads were base called using Dorado (v. 1.0.0) using super accuracy mode (model: r1041_e82_400bps_sup_v5.2.0). Long-read sequencing was performed at the Joint Microbiome Facility of the Medical University of Vienna and the University of Vienna under project ID JMF-2507-23.

## Bioinformatic analysis

An overview of the bioinformatic workflow is provided in Extended Data Fig. 2. All steps are described in detail in the following sections.

## Reference genomes

The genomes of all strains used in the mock communities are hereon referred to as reference genomes. They have been deposited in a public repository and published previously[24,25]. A phylogenetic tree was constructed from the genomes of the 70 isolates using PhyloPhlAn[27] v3.0.67 (options: --diversity medium --f supermatrix_aa.cfg) (Extended Data Fig. 1). Genome characteristics, including size and GC content, were analysed using Biopython[28] v.1.79 and bioawk[29] v1.0 (Supplementary Data 2).

## Preprocessing of shotgun metagenomic data

Raw reads of samples with the different empiric sequencing depths (Supplementary Data 3) were further subsampled bioinformatically to the exact targeted sequencing depth (below). The raw FASTQ files were subsampled using seqtk[30] v1.2 with default settings. The targeted number of paired-end reads (2 × 150 bp) per sequencing depth was as follows: 333,333 read pairs (0.10 Gb), 833,333 (0.25 Gb), 1,666,667 (0.5 Gb), 2,500,000 (0.75 Gb), 3,333,333 (1.0 Gb), 5,000,000 (1.5 Gb), 6,666,667 (2.0 Gb), 16,666,667 (5.0 Gb) and 33,333,333 (10.0 Gb). Adapters were removed and subsampled raw reads were quality-filtered with Trimmomatic[31] v.0.39 (options: TRAILING:3 LEADING:3 SLIDINGWINDOW:5:20 MINLEN:50 ILLUMINACLIP:{adapters.fa}:2:30:10). The bbduk command (options: hdist = 1 k = 31) in BBMap[32] v.38.84 was used for the removal of phiX sequences. Quality-filtered reads of all mock communities were assembled into contigs using MEGAHIT[33] v1.2.9. In addition, the quality-filtered reads of Mock-stag-70 were assembled into contigs with metaSPAdes v.4.2.0[34]. Contigs of metagenomes sequenced at 10 Gb, 20 Gb and 50 Gb assembled with either of the two assemblers were aligned to the reference genomes with blastn (perc_identity, 97%; evalue, 1 × 10$^{-10}$; alignment length, >150 bp). In addition to the three mock communities sequenced separately at all sequencing depths, and to the ten Mock-stag-24 communities with different compositions (v1-10) sequenced at 10 Gb, a further validation step included in silico subsampling of the 50-Gb Mock-stag-70 to 10 sequencing depth (0.1 Gb to 20 Gb), each ten times, using seqtk[30] v1.2 (default settings with ten different seeds).

## Taxonomic coverage

Coverage of the reference genomes by the quality-filtered metagenomic reads was determined using coverM[35] v0.6.1 (options: coverm genome--mapper bwa-mem--methods covered_fraction--min-covered-fraction 0--coupled). The relative abundance profiles were calculated using the read count option of coverM (options: coverm genome--methods count--min-covered-fraction 0--coupled), which were converted to relative abundance for each sample.

Non-supervised taxonomic profiles were generated using MetaPhlAn v4.0.1[10] with the mpa_vJun23_CHOCOPhlAnSGB_202403 database.

ANI values between the reference genomes of four *E. coli* strains and four *P. vulgatus* strains were calculated using FastANI[36] v1.34.

## Functional analysis

Protein-coding genes were predicted in the pooled reference genomes and the assembled contigs of the Mock samples (individually) using prodigal[37] v2.6.3.

Gene functions were annotated using KEGG Orthology and KofamScan[38] v1.3.0. Pathway completeness was assessed by KEGG-Decoder[12] v1.3, as the percentage of KEGG Ortholog covered, which are included in the manually curated canonical pathways used by KEGG-Decoder.

To determine the ability of each sequencing depth to capture the protein-encoding potential in the Mock communities, the genes predicted in the reference genomes were used to create a Diamond[39] (v2.0.15) protein sequence database, with which the protein sequences predicted in the assemblies (one assembly for each of the sequencing depths) were compared (options: diamond blastp --sensitive --query-cover 80 --id 90).

## Metagenome-assembled genomes

Contigs <1,000 bp were removed before reconstructing MAGs. An index table was built from the size-filtered contigs using bowtie2[40] v2.5.1 (bowtie2-build) with default options. The decontaminated paired-end reads were aligned to the bowtie index of size-filtered contigs (bowtie2 --S--very-sensitive-local --no-unal --p 30). Bam files were sorted with samtools[41] v1.17 (samtools view --bS). Contigs were binned using Metabat2[42] v2.12.1 and its algorithm for calculating coverage of each sequence in the assembly (jgi_summarize_bam_contig_depths) before creating the bins (metabat2 -m1500--maxP 95--minS 60--maxEdges 200--unbinned--seed 0).

The quality of the resulting bins was evaluated with checkM[43] v1.1.3 using the lineage workflow. Accordingly, the MAGs were categorized as being of high quality (>90% completeness, <5% contamination), medium quality (>70% completeness, <10% contamination) or low quality (all that did not fulfil the previous criteria). They were then taxonomically classified using GTDB-Tk[44] v2.3.2 (gtdbtk classify_wf) with the Genome Taxonomy Database r207. The coverage of the reference genomes by the different MAGs was determined using coverM[35] v0.6.1 (coverm genome--mapper bwa-mem--methods covered_fraction--min-covered-fraction 0 --single; no multiple read mapping). Reference genomes were aligned to MAGs by blastn v2.13.0 (-evalue 1e-10 -perc_identity 90.0), and circular alignments were plotted using shinyCircos[45].

For comparing sample-specific (approach above) with multicoverage binning for the ability to reconstruct high-quality, non-chimeric MAGs (that is, one genome match only), the workflow of Mattock and Watson[11] was used on Mock-stag-24 v1-10, sequenced at a depth of 10 Gb each.

## Long-read sequence analysis

The raw reads were filtered to a minimum average read quality of Q20 and a minimum length of 1,000 bp using chopper (v0.10.0)[46]. The filtered reads were used to randomly subset 11 additional datasets corresponding to 0.1, 0.25, 0.5, 0.75, 1, 1.5, 2, 5, 10, 20 and 50 Gb using rasusa (v2.1.1)[47]. Each subset was subsequently assembled using flye (v2.9.5)[48] with '--nano-hq' and polished once with medaka (v2.1.0, github.com/nanoporetech/medaka) using the –bacteria flag. Contigs <1,000 bp were removed. Assemblies were quality checked using QUAST[49] v5.3.0, and metagenomic binning was performed using metabat2[42]. Bin coverage was estimated using the jgi_summarize_bam_contig_depths function with a minimum percentage identity of 90. The quality and chimeric nature of MAGs was assessed as described above for the short-read data.

## Statistics and plotting

All statistical tests were performed in R (ref. 50) v4.4.1. using RStudio v2023.03.0+386 and the packages vegan[51], reshape[52], tidyverse[53] and dplyr[54]. For creation of the graphs, the R packages ggplot2[55] v3.5.1, ggpubr[56] v0.6.0, ggbreak[57] and ggfortify[58] v0.4.17 were used. The specific statistical tests used in each analysis are stated in the Results.

## Reporting summary

Further information on research design is available in the Nature Portfolio Reporting Summary linked to this article.

## Data availability

The raw metagenomic sequencing data was deposited at the European Nucleotide Archive/NCBI and are accessible under project no. PRJEB83573. All strains (and their genomes) used in this study are available via the Leibniz Institute DSMZ (German Collection of Microorganisms and Cell Cultures) at http://www.dsmz.de/miBC and https://hibc.rwth-aachen.de/. Source data are provided with this paper.

## Code availability

The code used for bioinformatic analyses is available via GitHub at https://github.com/ClavelLab/Benchmarking-shallow-Metagenomics.

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

## Acknowledgements

We are grateful to E. Deis for sample processing, A. Viehof-Beckmann for providing gut content of germ-free mice, and N. Kousetzi for outstanding support with molecular work, and C. Driessen for genomic library preparation. This work was supported by the Genomics Facility of the Interdisciplinary Center for Clinical Research (IZKF) Aachen within the Faculty of Medicine at RWTH Aachen University; the DFG-funded NGS Competence Center Tübingen (INST 37/1049-1) and the Institute for Medical Microbiology and Hygiene at the University Hospital (Tübingen, Germany), including help by the Quantitative Biology Center (QBiC) for raw data management and storage; and the Life Science Computer Cluster (LiSC) of the University of Vienna for the processing of long-read sequencing data. T.C. received funding from the German Research Foundation (DFG), project no. 403224013 (SFB1382, Q02), project no. 460129525 (NFDI4Microbiota) and project no. 445552570. D.B. received funding from the Austrian Science Fund (10.55776/DOC69; 10.55776/COE7).

## Author contributions

N.T.: conceptualization, methods and software, investigation, formal analysis, validation, interpretation, visualization, data curation, writing – original draft; C.P.: methods and software, validation, interpretation, writing – review and editing; J.S. and P.P.: investigation, formal analysis, writing – review and editing; D.B. and J.P.: resources, funding acquisition, writing – review and editing; T.C.A.H.: conceptualization, interpretation, writing – original draft; T.C.: conceptualization, visualization, interpretation, resources, supervision, project administration, funding acquisition, writing – original draft.

## Funding

## Competing interests

The authors declare no competing interests.

## Additional information

**Extended data** is available for this paper at https://doi.org/10.1038/s41564-026-02334-2.

**Correspondence and requests for materials** should be addressed to Nicole S. Treichel or Thomas Clavel.

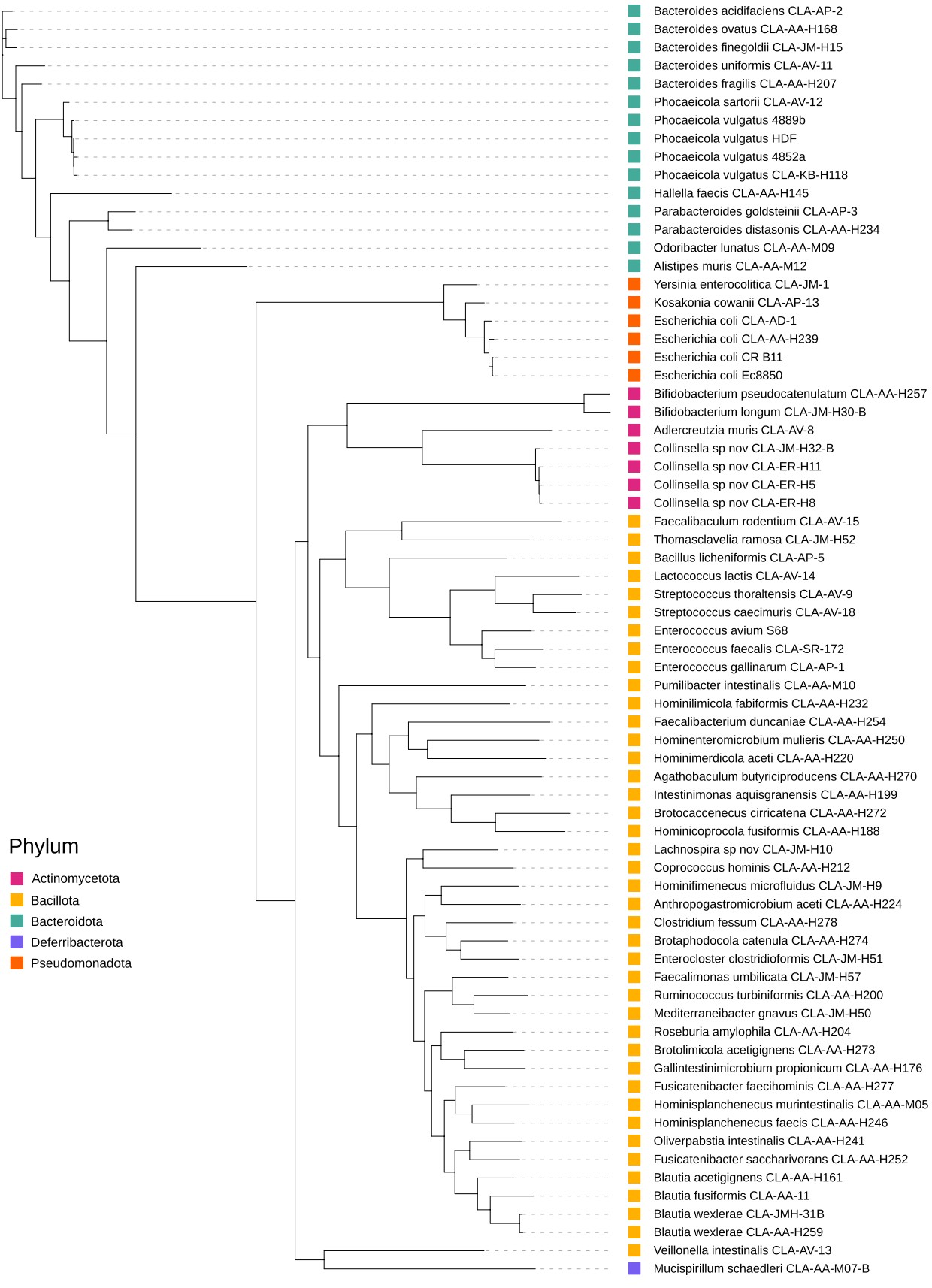

**Extended Data Fig. 1 | Phylogenetic tree of bacterial strains in the mock communities.** A phylogenetic tree of high-quality, draft genomes of the 70 isolates was created using PhyloPhlAn v3.0.67. The phyla to which the strains belong are represented by coloured boxes at the end of the branches. Distribution of the strains in each mock community is provided in Supplementary Data 1.

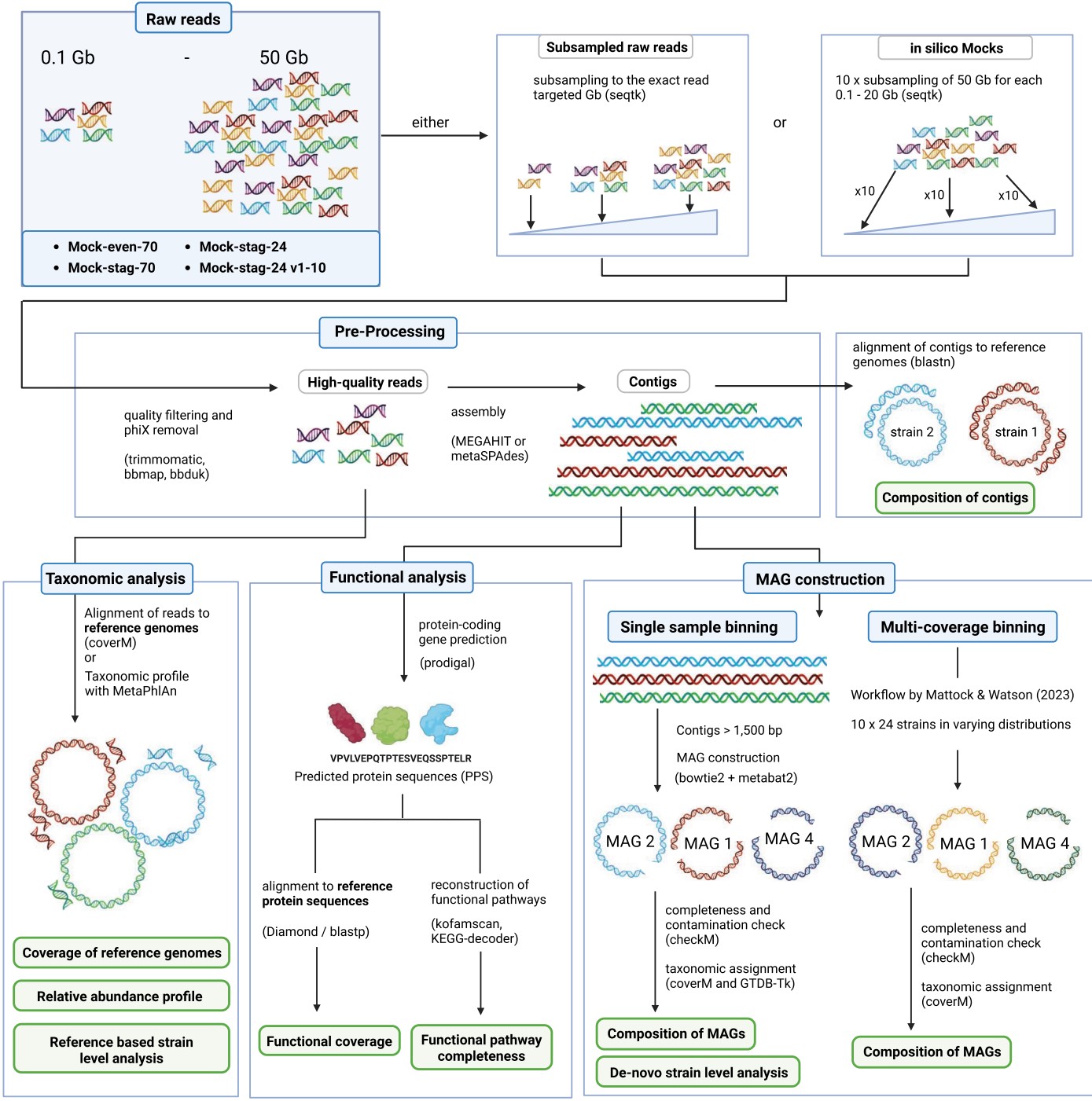

**Extended Data Fig. 2 | Schematics of the bioinformatics workflow.** The details of each analysis are described in the methods. (Created with BioRender). Figure created in BioRender; Treichel, N. https://biorender.com/63tt9vp (2026).

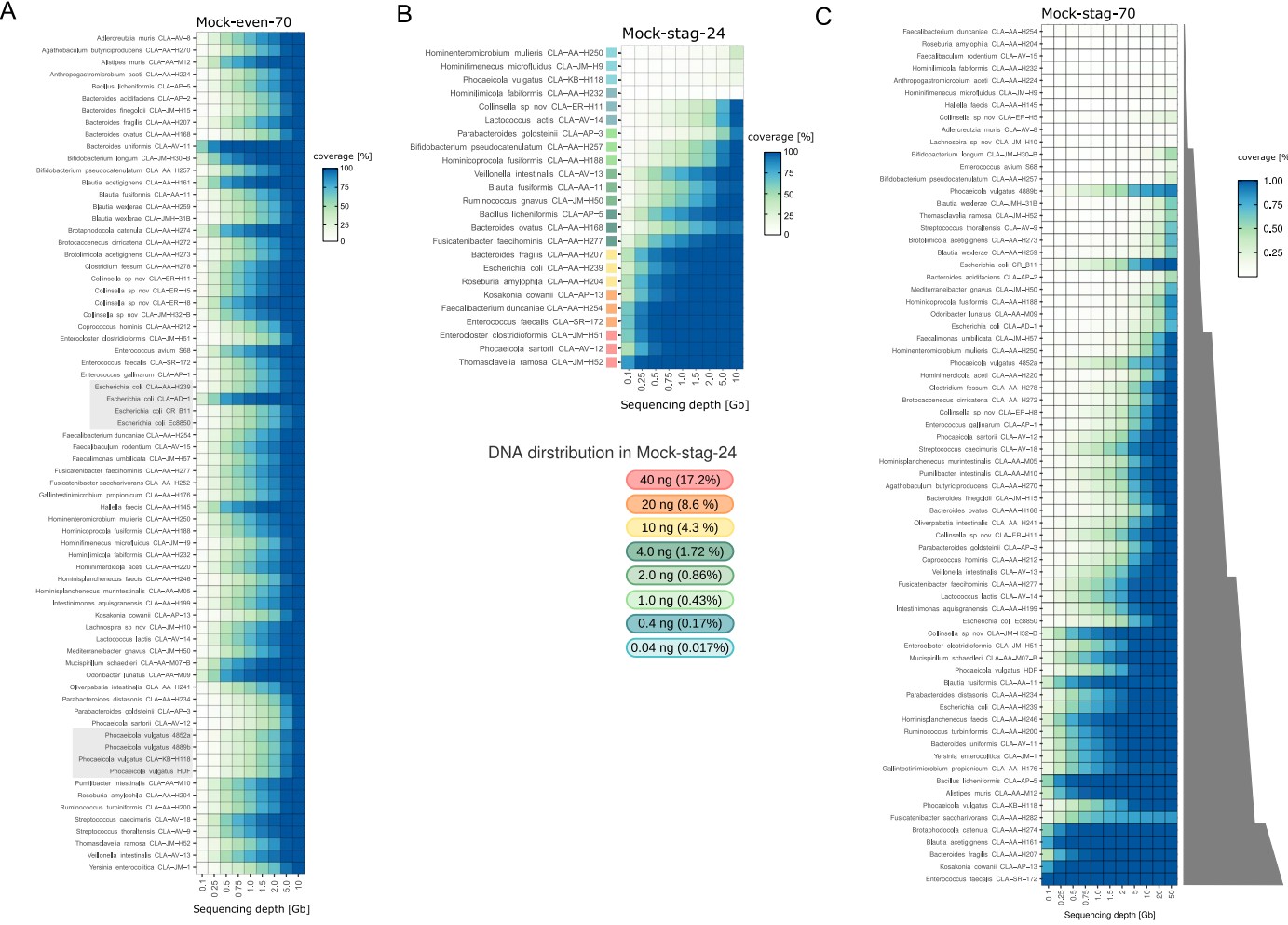

**Extended Data Fig. 3 | Coverage of individual reference genomes.** Heatmaps showing the coverage of all reference genomes in Mock-even-70 (**A**), Mock-stag-24 (**B**), and Mock-stag-70 (**C**) by metagenomic reads at up to 11 sequencing depths (x-axis). The grey boxes in panel **A** indicate the multiple *E. coli* and *P. vulgatus* strains. The reference genomes in the mock communities with staggered distribution in panel **B** and **C** were ranked (from top to bottom) according to increasing DNA amount in the mixture, as indicated by the colour gradient in panel B (from blue to red; 0.04, 0.4, 1, 2, 4, 10, 20, 40 ng) or the grey gradient in panel C (concentrations are provided in Supplementary Data 1).

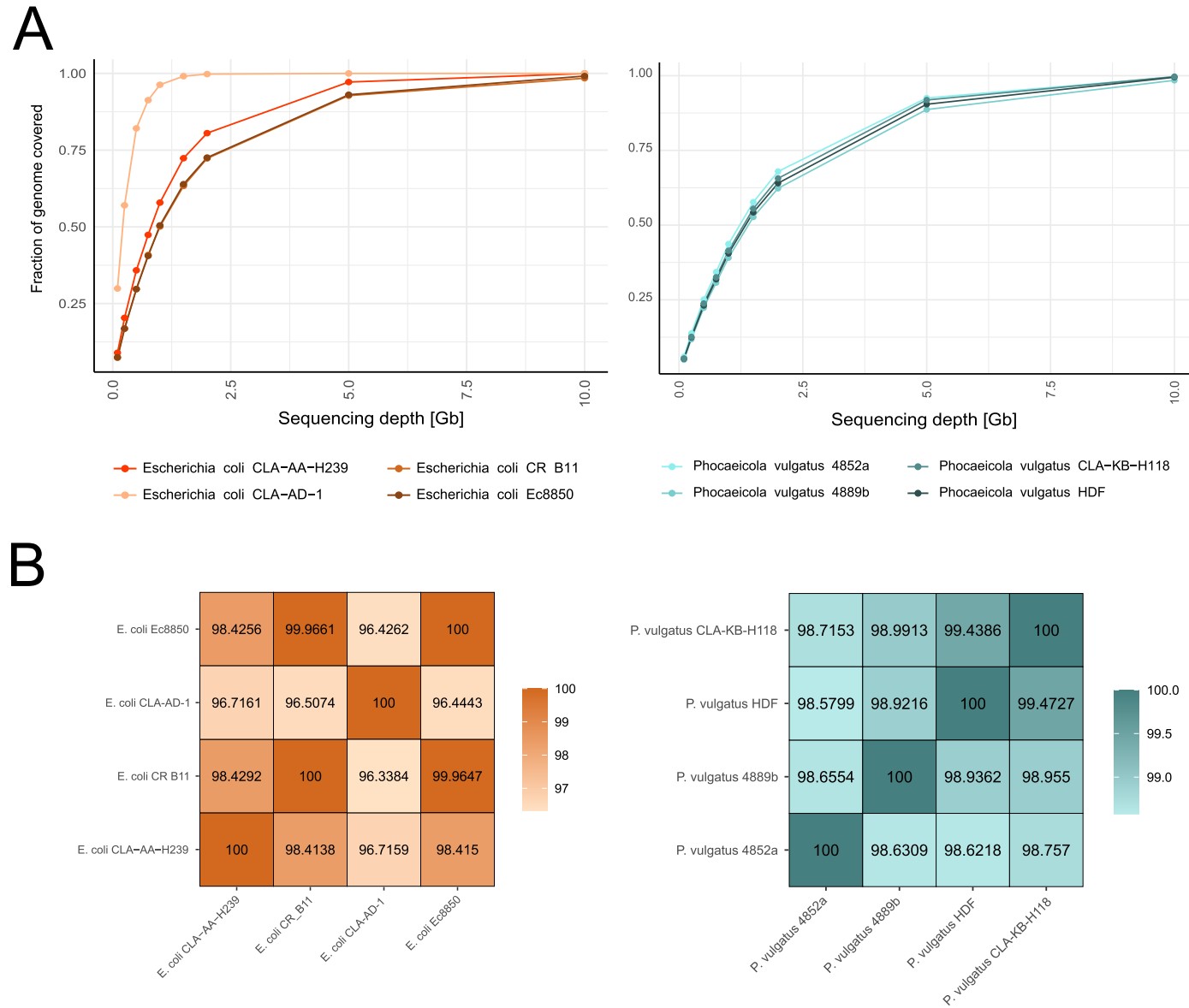

**Extended Data Fig. 4 | Strain level resolution based on reference genomes. A** Reference genome coverage (y-axis) for the four strains of *E. coli* and *P. vulgatus* at the nine different sequencing depths (x-axis). **B** ANI values between the strains within each species.

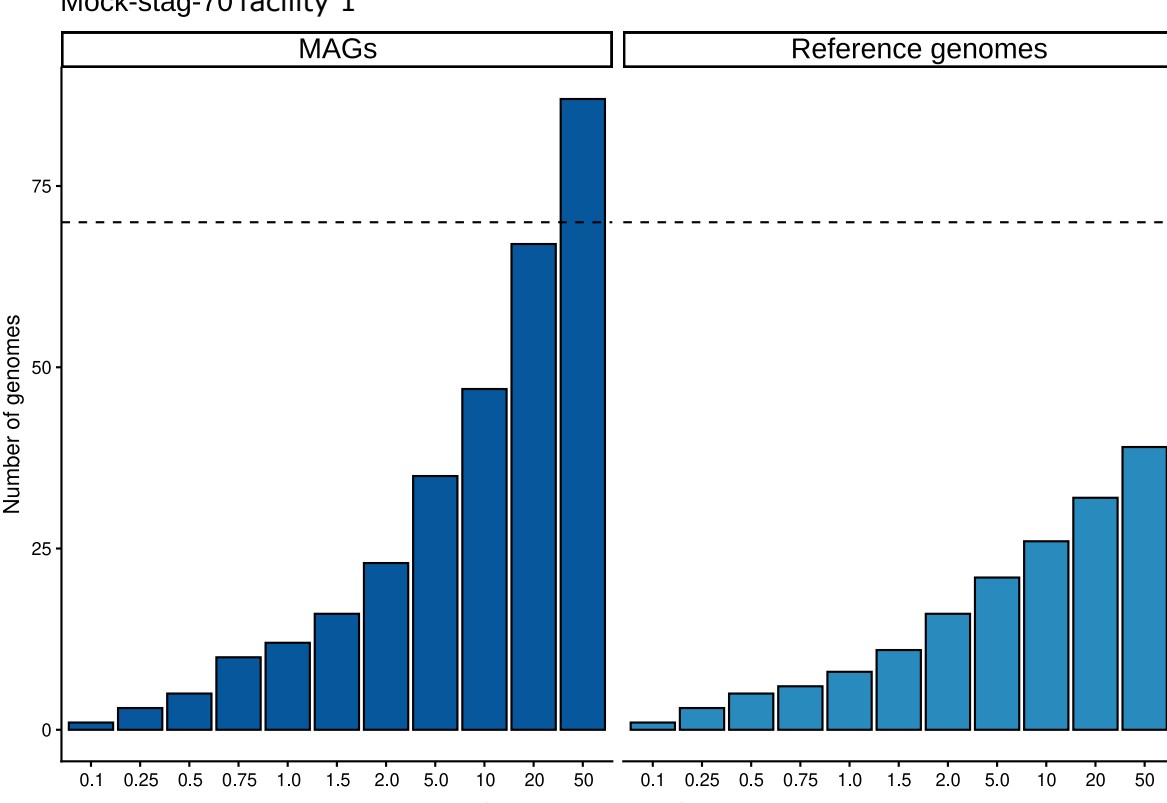

**Extended Data Fig. 5 | MAGs numbers in Mock-stag-70.** Strain analysis based on the MAGs assembled in Mock-stag-70 using MEGAHIT for assembly at each sequencing depth individually. The left panel (dark blue) depict the number of MAGs assembled from the shotgun data; the right panel (lighter blue) show the number of reference genomes matching MAGs (coverM), indicating that multiple MAGs were reconstructed for some Mock species. The reference genome with the highest coverage by MAG reads was chosen as representative of that MAG.

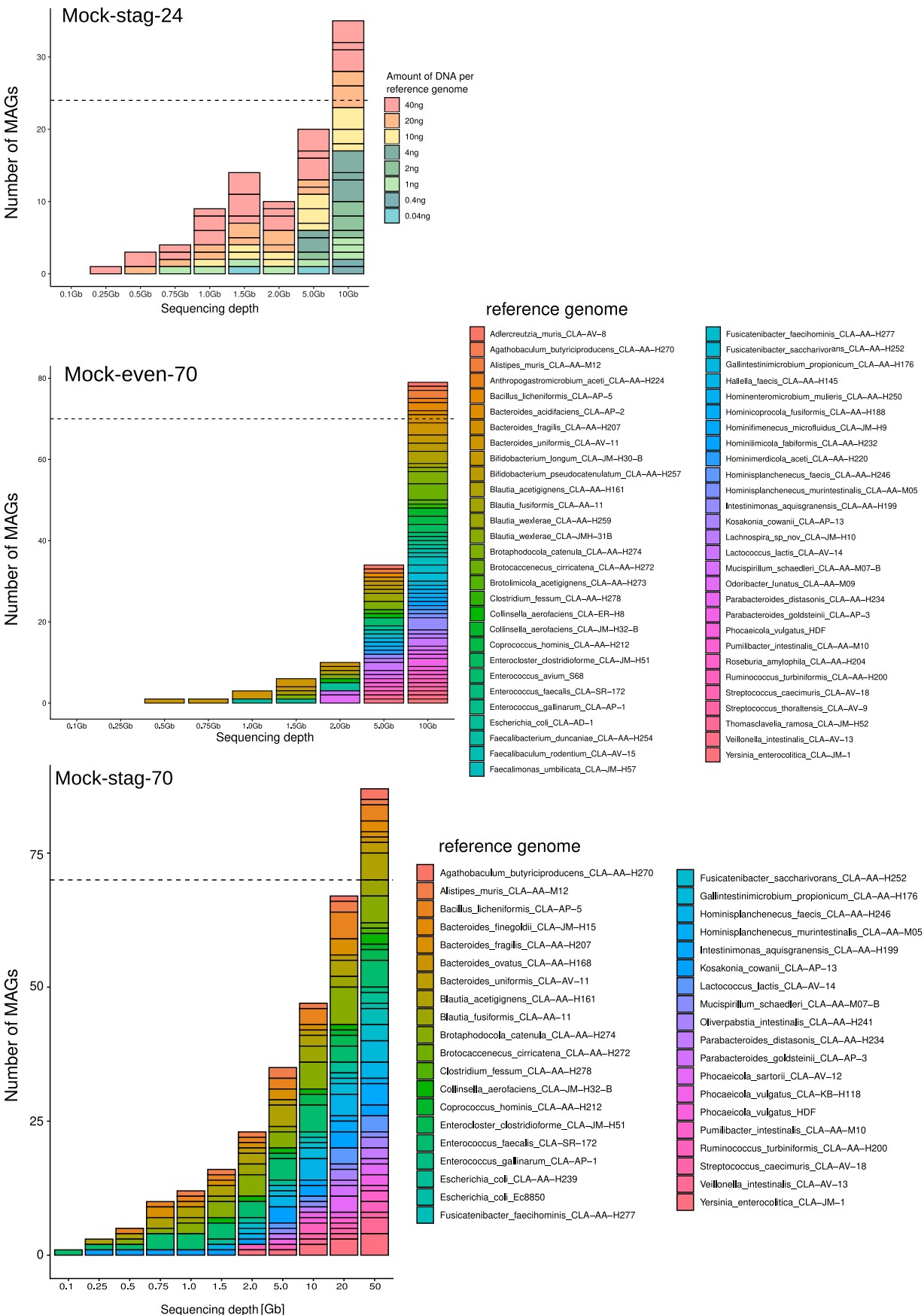

**Extended Data Fig. 6 | Reference genomes per MAG for Mock-stag-24, Mock-even-70, and Mock-stag-70.** Stacked bar charts of the number of MAGs per sequencing depth. Each box represents one reference genome. The size of the box indicates the number of MAGs that were assigned to this reference genome. The colours in the graph for Mock-stag-24 indicate the amount of input DNA for a given strain in the pool. The colour of the graph for Mock-even-70 and Mock-stag-70 indicates the reference genomes.

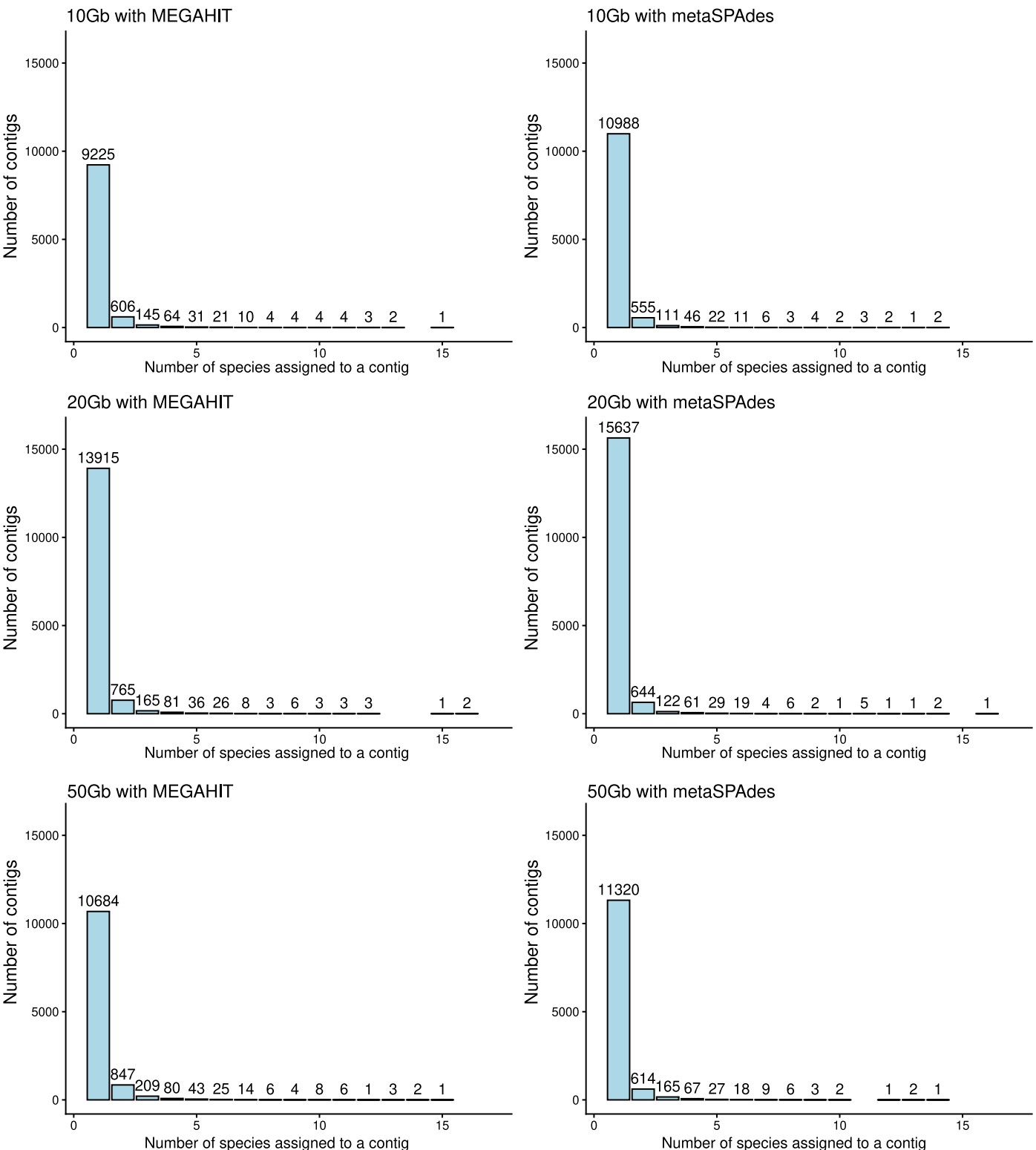

**Extended Data Fig. 7 | Contigs assigned to reference genomes.** Contigs from metagenomes sequenced at 10 Gb, 20 Gb and 50 Gb either assembled with MEGAHIT or metaSPAdes were aligned to the reference genomes with blastn (-perc_identity 97% -evalue 1e-10, alignment length >150 bp). The number of contigs (y-axis) is plotted against the number of reference genomes assigned to a contig (x-axis).

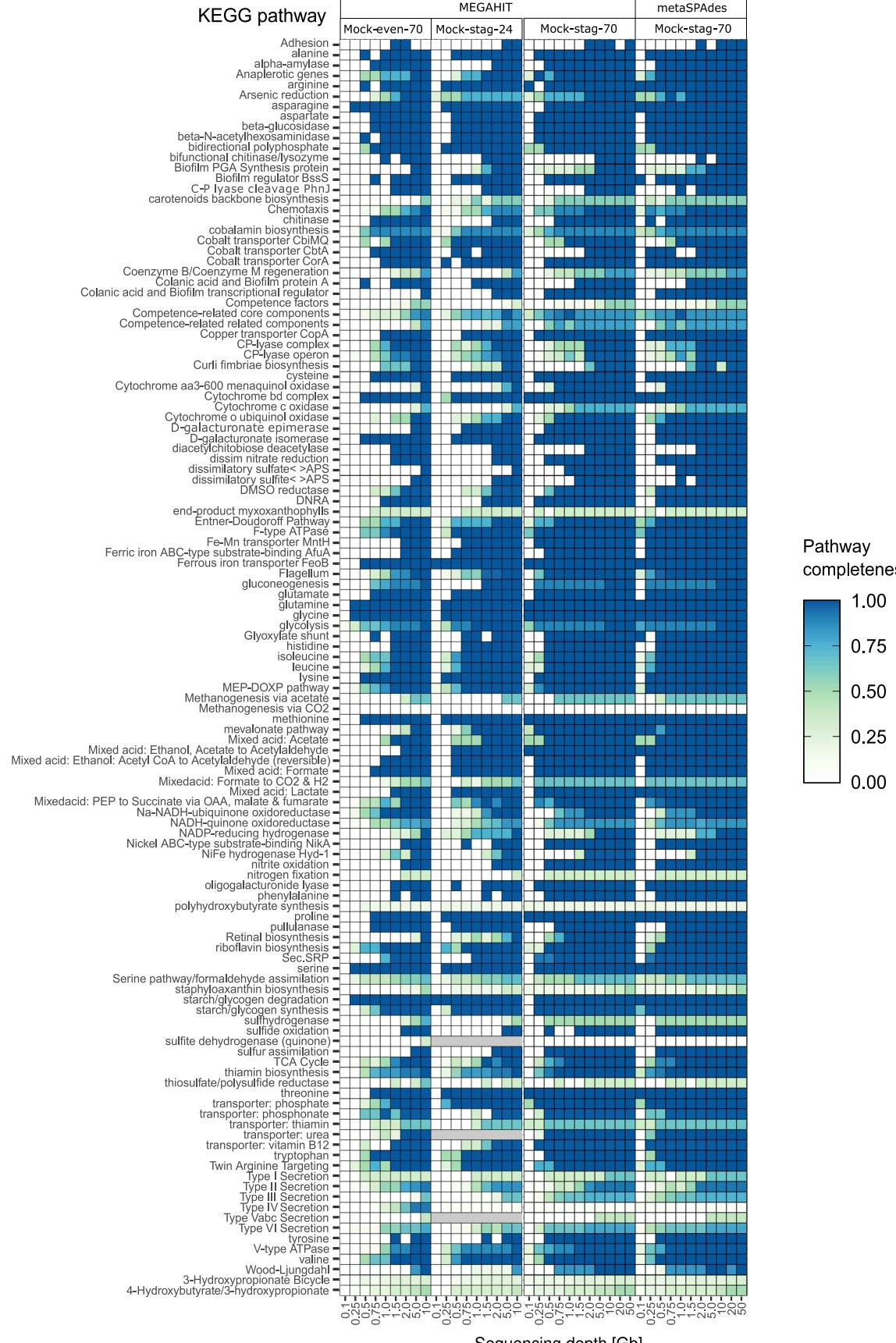

**Extended Data Fig. 8 | Heatmaps of functional coverage.** Completeness of all KEGG pathways present in the reference genomes in Mock-even-70, Mock-stag-24 and Mock-stag-70 assembled with MEGAHIT, and Mock-stag-70 assembled with metaSPAdes (from left to right) at the up to 11 sequencing depths (columns).

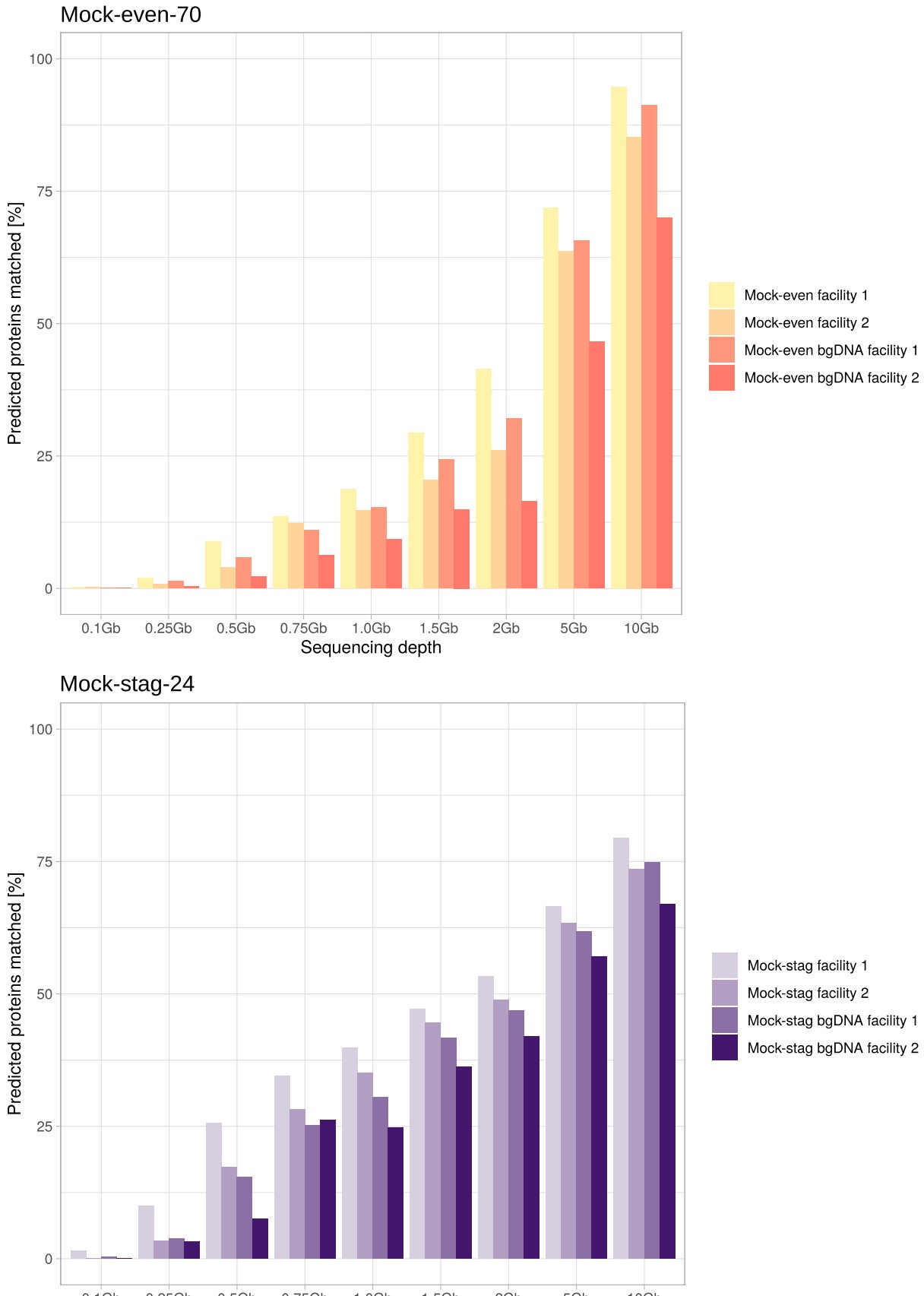

**Extended Data Fig. 9 | Bar charts of protein sequences recovered.** Protein sequenced predicted in the metagenomic datasets as percentage of the protein sequences present in the reference genomes (y-axis). The colour gradient indicates the facility where the library was prepared and the addition of background DNA (bgDNA) from the caecum of germfree mice.

# Reporting Summary

## Statistics

For all statistical analyses, confirm that the following items are present in the figure legend, table legend, main text, or Methods section.

| n/a | Confirmed | |
|---|---|---|
| ☐ | ☒ | The exact sample size (*n*) for each experimental group/condition, given as a discrete number and unit of measurement |
| ☐ | ☒ | A statement on whether measurements were taken from distinct samples or whether the same sample was measured repeatedly |
| ☐ | ☒ | The statistical test(s) used AND whether they are one- or two-sided<br>*Only common tests should be described solely by name; describe more complex techniques in the Methods section.* |
| ☒ | ☐ | A description of all covariates tested |
| ☐ | ☒ | A description of any assumptions or corrections, such as tests of normality and adjustment for multiple comparisons |
| ☐ | ☒ | A full description of the statistical parameters including central tendency (e.g. means) or other basic estimates (e.g. regression coefficient) AND variation (e.g. standard deviation) or associated estimates of uncertainty (e.g. confidence intervals) |
| ☐ | ☒ | For null hypothesis testing, the test statistic (e.g. *F*, *t*, *r*) with confidence intervals, effect sizes, degrees of freedom and *P* value noted<br>*Give P values as exact values whenever suitable.* |
| ☒ | ☐ | For Bayesian analysis, information on the choice of priors and Markov chain Monte Carlo settings |
| ☒ | ☐ | For hierarchical and complex designs, identification of the appropriate level for tests and full reporting of outcomes |
| ☒ | ☐ | Estimates of effect sizes (e.g. Cohen's *d*, Pearson's *r*), indicating how they were calculated |

*Our web collection on statistics for biologists contains articles on many of the points above.*

## Software and code

Policy information about availability of computer code

| | |
|---|---|
| Data collection | Illumina NovaSeq6000 control software v.1.8.1; Oxford Nanopore Technologies Minknow v. 25.05.12 |
| Data analysis | biobython v.1.7, bioawk v1.0, seqtk v1.2, Trimmomatic v.0.39, BBMap v.38.84, MegaHit v1.2.9, coverM v0.6.1, FastANI v1.34, prodigal v2.6.3, KofamScan v1.3.0, KEGG-Decoder v1.3, Diamond v2.0.15, bowtie2 v2.5.1, samtools v1.17, Metabat2 v2.12.1, checkM v1.1.3, GTDB-Tk v2.3.2, blastn v2.13.0, shinyCircos, R v4.4.1., RStudio v2023.03.0+386, Dorado v. 1.0.0, MetaPhlAn v4.0.1, chopper v0.10.0, rasusa v2.1.1, flye v2.9.5, medaka v2.1.0, QUAST v5.3.0<br>https://github.com/ClavelLab/Benchmarking-shallow-Metagenomics |

For manuscripts utilizing custom algorithms or software that are central to the research but not yet described in published literature, software must be made available to editors and reviewers. We strongly encourage code deposition in a community repository (e.g. GitHub). See the Nature Portfolio guidelines for submitting code & software for further information.

## Data

Policy information about availability of data

All manuscripts must include a data availability statement. This statement should provide the following information, where applicable:

- Accession codes, unique identifiers, or web links for publicly available datasets
- A description of any restrictions on data availability
- For clinical datasets or third party data, please ensure that the statement adheres to our policy

> The raw metagenomic sequencing data was deposited at the European Nucleotide Archive/NCBI and is accessible under Project no. PRJEB83573

## Research involving human participants, their data, or biological material

Policy information about studies with human participants or human data. See also policy information about sex, gender (identity/presentation), and sexual orientation and race, ethnicity and racism.

| | |
|---|---|
| Reporting on sex and gender | *Use the terms sex (biological attribute) and gender (shaped by social and cultural circumstances) carefully in order to avoid confusing both terms. Indicate if findings apply to only one sex or gender; describe whether sex and gender were considered in study design; whether sex and/or gender was determined based on self-reporting or assigned and methods used. Provide in the source data disaggregated sex and gender data, where this information has been collected, and if consent has been obtained for sharing of individual-level data; provide overall numbers in this Reporting Summary. Please state if this information has not been collected. Report sex- and gender-based analyses where performed, justify reasons for lack of sex- and gender-based analysis.* |
| Reporting on race, ethnicity, or other socially relevant groupings | *Please specify the socially constructed or socially relevant categorization variable(s) used in your manuscript and explain why they were used. Please note that such variables should not be used as proxies for other socially constructed/relevant variables (for example, race or ethnicity should not be used as a proxy for socioeconomic status). Provide clear definitions of the relevant terms used, how they were provided (by the participants/respondents, the researchers, or third parties), and the method(s) used to classify people into the different categories (e.g. self-report, census or administrative data, social media data, etc.) Please provide details about how you controlled for confounding variables in your analyses.* |
| Population characteristics | *Describe the covariate-relevant population characteristics of the human research participants (e.g. age, genotypic information, past and current diagnosis and treatment categories). If you filled out the behavioural & social sciences study design questions and have nothing to add here, write "See above."* |
| Recruitment | *Describe how participants were recruited. Outline any potential self-selection bias or other biases that may be present and how these are likely to impact results.* |
| Ethics oversight | *Identify the organization(s) that approved the study protocol.* |

Note that full information on the approval of the study protocol must also be provided in the manuscript.

# Field-specific reporting

Please select the one below that is the best fit for your research. If you are not sure, read the appropriate sections before making your selection.

☒ Life sciences  ☐ Behavioural & social sciences  ☐ Ecological, evolutionary & environmental sciences

For a reference copy of the document with all sections, see nature.com/documents/nr-reporting-summary-flat.pdf

# Life sciences study design

All studies must disclose on these points even when the disclosure is negative.

| | |
|---|---|
| Sample size | No sample size calculation was perfomed |
| Data exclusions | No data was excluded |
| Replication | Mock communities of several sizes and compositions were included. Additionally in-silico subsampling was perfomed to generate replicates (n=10) for several analyses. |
| Randomization | N.A. |
| Blinding | N.A. |

# Reporting for specific materials, systems and methods

We require information from authors about some types of materials, experimental systems and methods used in many studies. Here, indicate whether each material, system or method listed is relevant to your study. If you are not sure if a list item applies to your research, read the appropriate section before selecting a response.

## Materials & experimental systems

| n/a | Involved in the study |
|-----|-----------------------|
| ☒ | ☐ Antibodies |
| ☒ | ☐ Eukaryotic cell lines |
| ☒ | ☐ Palaeontology and archaeology |
| ☐ | ☒ Animals and other organisms |
| ☒ | ☐ Clinical data |
| ☒ | ☐ Dual use research of concern |
| ☒ | ☐ Plants |

## Methods

| n/a | Involved in the study |
|-----|-----------------------|
| ☒ | ☐ ChIP-seq |
| ☒ | ☐ Flow cytometry |
| ☒ | ☐ MRI-based neuroimaging |

## Animals and other research organisms

Policy information about [studies involving animals](); [ARRIVE guidelines]() recommended for reporting animal research, and [Sex and Gender in Research]()

| | |
|---|---|
| Laboratory animals | Mus musculus; C57BL/6 germfree |
| Wild animals | *Provide details on animals observed in or captured in the field; report species and age where possible. Describe how animals were caught and transported and what happened to captive animals after the study (if killed, explain why and describe method; if released, say where and when) OR state that the study did not involve wild animals.* |
| Reporting on sex | Caecal content of both male and female mice was used. |
| Field-collected samples | *For laboratory work with field-collected samples, describe all relevant parameters such as housing, maintenance, temperature, photoperiod and end-of-experiment protocol OR state that the study did not involve samples collected from the field.* |
| Ethics oversight | Institute of Laboratory Animal Science at the University Hospital of RWTH Aachen under Ethical Approval (LANUV no. 81-02.04.2023.A253) and in accordance with the German Animal Protection Law (TierSchG). |

Note that full information on the approval of the study protocol must also be provided in the manuscript.

## Plants

| | |
|---|---|
| Seed stocks | *Report on the source of all seed stocks or other plant material used. If applicable, state the seed stock centre and catalogue number. If plant specimens were collected from the field, describe the collection location, date and sampling procedures.* |
| Novel plant genotypes | *Describe the methods by which all novel plant genotypes were produced. This includes those generated by transgenic approaches, gene editing, chemical/radiation-based mutagenesis and hybridization. For transgenic lines, describe the transformation method, the number of independent lines analyzed and the generation upon which experiments were performed. For gene-edited lines, describe the editor used, the endogenous sequence targeted for editing, the targeting guide RNA sequence (if applicable) and how the editor was applied.* |
| Authentication | *Describe any authentication procedures for each seed stock used or novel genotype generated. Describe any experiments used to assess the effect of a mutation and, where applicable, how potential secondary effects (e.g. second site T-DNA insertions, mosiacism, off-target gene editing) were examined.* |

