## [Peer Review File · Nature Microbiology]

Benchmarking of shotgun sequencing depth reveals the potential and limitations of shallow metagenomics and strain-level analysis

Corresponding Author: Professor Thomas Clavel

Version 0:

Decision Letter:

5th February 2025

Dear Tom

Thank you very much for your enquiry about submitting a manuscript to Nature Microbiology.

I've now had a chance to discuss your work with my colleagues. We think that it sounds very interesting and important.

Therefore, we would like to invite you to submit the full manuscript to Nature Microbiology so that we can examine the data before deciding whether to send the paper out to review.

If this is acceptable to you, you can submit the complete manuscript using the link below:

Link Redacted

If you have any questions, please feel free to contact me.

Yours sincerely

Version 1:

Reviewer comments:

Reviewer #1

(Remarks to the Author)

This manuscript addresses some very important questions about the accuracy, reproducibility and interpretability of metagenome datasets. Many metagenomic studies generate, analyze and interpret metagenome data without any significant effort to validate or ground-truth the methods use, and only a small number of studies have aimed to benchmark metagenomic workflows against a "gold standard". This is challenging to do, since the true nature of "real" communities is unknown and synthetic communities fail to reproduce the complexity of natural communities, particularly in terms of strain heterogeneity. Synthetic data, also often used for this type of study, fails to reproduce the potential biases resulting from the library construction and sequencing process. Some of these weaknesses are addressed in the current study.

Although the authors introduce the study with a focus on the utility of shallow metagenome sequencing, and the depth necessary to gain phylogenetic and functional insights, their most notable findings relate to deep metagenomics – in particular the sometimes very poor quality of "high quality" metagenome assembled genomes. Overall, they find that shallow sequencing is effective for phylogenetic and functional characterization only when a high-quality reference database is available, but that such a high-quality database cannot be generated based on MAGs from deep sequencing. This is based on the finding that MAGs, even those derived from deep sequencing that appear high quality based on standard metrics, are frequently chimeric and/or redundant and do not accurately represent the functional gene complement of the source genomes.

While this finding is notable, it is unfortunately based on a relatively limited data set that is not fully explored. While the authors stress the weaknesses of other benchmarking studies based on deep sequencing of native communities, bioinformatic subsampling of data and/or in silico generation of mock communities, the specific details of their protocols for generating datasets of different depths are glossed over in multiple places. It appears that different libraries were constructed for each target

sequencing depth, and then “pooled to reflect the nine different sequencing depths targeted” – it would have been helpful to see more about how this was done, since the results are presented as very precise basecounts per library (e.g. 0.1 Gb, 0.25 Gb, etc.) – did the pooling strategy really yield exactly the sequence amounts desired, or was there some bioinformatic subsampling? More importantly, was there really only one library made per target depth with no replication? For a benchmarking study, wouldn't proper replication and statistics be critical to any findings? The resultant lack of statistical significance for any of the findings makes essentially all of the observations anecdotal.

Another key weakness is the use of just one sequence assembly method, MegaHIT. While a comprehensive analysis of different assemblers would be a major effort in itself, no justification is given for choosing this particular assembler over many others that could potentially produce higher quality and less chimeric MAGs. Without further investigation of this question, the assertion that MAGs fail to accurately represent the source genomes is unwarranted.

I also thought the authors downplayed the contributions of previous studies, such as Bowers et al 2015 (reference 43) which also made libraries from mixed isolate DNA for metagenome sequencing but was only briefly mentioned in the discussion. There is also a large metagenome benchmarking effort called the Critical Assessment of Metagenome Interpretation (CAMI; <https://camichallenge.org/>) which is more focused on bioinformatics tools but addresses similar questions and does not appear to be referenced at all.

Overall, I appreciate the effort to ground truth common metagenomic approaches, and the recognition that investigators may put too much faith in the tools they use. But this limited study only provides a glimpse at how things can go wrong and doesn't offer any novel solutions, or guidance that extends much beyond the specific communities examined.

A few more specific comments on the manuscript are below.

Line 43: “Shallow metagenomics, defined as shotgun sequencing at a depth ≤ 1 Gb (ca. 3 million reads)...” Where does this definition come from?

Line 94: “Fragment size of cleaned DNA was determined on an Agilent D1000 TapeStation...” What was the fragment size of these libraries?

Line 288: I'm not sure an equation is needed for this simple arithmetic. However, if the insert size is 250 bp, doesn't this equation overestimate the amount of unique sequence generated by treating each 150 bp read as representing unique sequence?

Figure 2A: The colors for strain CLA-AA-H239 and CR B11 are nearly identical.

Line 361: “However, sequencing depths > 5 Gb are required to obtain nearly full genome coverage...” This seems wholly dependent on the composition of the community. A similar statement is made on line 382 (“...showing that > 2 Gb is required to reach $> 50\%$ completeness.”

Lines 374-380: This section suggests potential problems with binning and ways to improve, but glaringly ignores potential assembly errors.

Lines 44, 79, 273: There should be no comma after “both”

Line 281: “KEEG” should be “KEGG”

Reviewer #2

(Remarks to the Author)

The manuscript by Treichel NS, et al., presents the impact of sequencing depth on metagenomic analysis. The authors used two mock bacterial DNA communities (even and staggered distributions) and analyzed them at nine different sequencing depths (from 0.1 Gb to 10.0 Gb). The study also explored the influence of different library preparation protocols and the presence of host DNA on the results. The key findings highlight that while reference-based taxonomic and strain-level insights can be retrieved at relatively shallow sequencing depths (0.5-1.0 Gb), de novo metagenome-assembled genome (MAG) reconstruction requires significantly deeper sequencing (> 10 Gb) and often results in chimeric MAGs. Functional pathway analysis was found to be reliable at 2 Gb, but comprehensive proteome coverage required 10 Gb of sequencing data.

Major points

The study evaluates the impact of actual variations in sequencing depth, which is highly valuable for practical experimental design. As a complementary perspective, the authors might consider discussing or, if feasible, briefly exploring the use of technical subsampling (rarefaction). By taking the deepest sequenced libraries and computationally subsampling reads to simulate shallower depths, it could help further disentangle limitations arising purely from statistical effects of reduced read counts versus potential biases introduced during library preparation or the sequencing process itself when targeting lower depths from the outset. This could add another dimension to the interpretation of why certain analytical goals are challenging at shallower depths.

Staggered community: What is the distribution? (e.g., log-normal? Or others?)

Line 56-58, downsampling sequencing of higher depth vs. directly sequencing at lower depth – citation or comparison here?

L176-182, the authors used coverM with bwa-mem mapper. Since the reference genomes are sometimes very similar (different strains of the same species), are multi-mapping reads showing up, or is every read uniquely assigned to the correct reference? If multi-mapping reads exist, how are these handled, and how does it affect the coverage calculation? Can it explain cases with very low coverage, and that almost all genomes never achieve 100% coverage at any depth? Also related to analysis at L217-223, for genome pairs with higher ANI, multi-mapping reads could be more likely.

L200-204 Are the outlier relative abundances due to imprecision in the lab or due to some computational artifact?

L234 "This suggests MAGs become increasingly fragmented with greater sequencing depth, rather than coalescing into a single high-quality MAG". This statement doesn't immediately follow from the data in Fig. 2C, as both the number of MAGs and the number of references covered are increasing. The fragmentation appeared not to be assessed. A more in-depth discussion or hypothesis as to why this occurs would also strengthen the paper. Could this be related to the assembly algorithms used or inherent complexities in resolving highly similar genomes?

L244-249. Are the different contigs of the final MAG coming from different strains? According to the Supp. Table 3, they were assigned to multiple ref. genomes. Is it possible that the contigs themselves were assembled from reads coming from multiple strains? There is a whole class of assembly and binning tools that attempt "strain-aware" metagenomic assembly, while the tools used by the authors don't consider such cases, and so a single bin per species, possibly a strain hybrid/chimera, is expected. Maybe the authors could at least comment on this in the manuscript.

L250-263, Fig2D. The authors use checkM to estimate the completeness and contamination of the MAGs, but having the ground truth genomes available and having done the mapping, the authors could provide more exact values of true completeness (coverage of the assigned reference) and contamination (coverage of other references). The distinction of whether the MAG was assigned to one or multiple references is not very meaningful, as this doesn't reflect the size of the portion of the MAG coming from other references – the contamination in the checkM sense of meaning. I understand it's not straightforward to visualize in one plot, but having shown how with increasing seq depth the number of MAGs with different true completeness and true contamination changes would be more illustrative.

Moreover, there is a drop in the number of MAGs from 5Gb to 10Gb in the staggered case, which could support the previous statement on L334 - with higher depth, the tool tends to over-bin the contigs. Has this been reported previously in the literature? Is this specific to the binner used by the authors? As a simpler analysis preceding the MAGs, the authors could consider working just with contigs without binning and assessing how much the reference genomes are covered by them at different depths, as the chimeric MAGs are a limitation of the binner.

L265-265 "de-novo analysis based on MAG reconstruction leads to contaminated genomes, even at supposed high-quality". The high-quality definition allows for contamination up to 5%, the authors could show how much of the true contamination there is based on the coverage of ref. genomes, but it would be just performing a benchmark of checkM prediction accuracy. Instead, the authors could investigate whether higher sequencing depth and thus higher coverage of certain genomes will lead to lower contamination or not.

L266-268 "Together with the creation of multiple MAGs for individual strains, these findings suggest that strain-level diversity within MAG catalogues is artificially inflated". Does it happen that one ref. genome would be represented by multiple MAGs that are of medium or high completeness, and thus they overlap? Or is it that their contigs get split into multiple disjoint sets? To support the statement of the strain-level diversity being inflated, the authors would have to show that a single strain could be assembled and binned to multiple MAGs that overlap each other, and moreover they would pass certain de-replication thresholds used by MAG catalogues. I believe this analysis would be of interest to the readers.

L342-363 The authors should highlight the limitation of their analysis, that is, that they always mapped the reads against the reference genomes of the exact strains they were looking for. This might be similar to some applications of pathogen detection where the exact strain is known, but for general microbiome analyses, usually only the ref. genomes of other strains of the species are available, and genomes of all possible species are included in the database. This might reduce both the sensitivity and also introduce false positives, leading to a lower signal-to-noise ratio in the lower seq depths than the ideal case presented here. It should also be mentioned that k-mer or marker gene-based approaches will have different merits and limitations compared to the full genome mapping considered here.

The authors show that library preparation and presence of background DNA significantly impact the results, especially at lower seq depths. It would be nice to discuss more about the influence of these factors, for example, does lower template DNA or more PCR cycles specifically lead to biases or reduced coverage? The manuscript mentions that Facility 1, using more template DNA and fewer PCR cycles, yielded more robust profiles. It would be valuable to discuss whether differences between facilities could also be due to other factors not taken into account (e.g., bead cleanup?).

Minor points

The title should include "shallow sequencing" since that is the target application of the benchmark

L61: "Given the urgent need to better define the strengths and limitations of shallow metagenomics, we systematically evaluated..." This sentence appears twice consecutively.

L111: "biobytion13" should likely be "Biopython".

L185-186: "Thomasclavelia ramosa CLA-JM-H52 and Enterocloster clostridioformis CLA-JM-H51, which were added to the Mock with highest DNA concentration (17.2%; 40 ng), and V. intestinalis CLA-AV-13, added at a low concentration (0.86%, 2 ng)." Ensure clarity that V. intestinalis reaching 100% coverage despite low input is a notable point, perhaps linked to its genome characteristics or sequencing efficiency.

L221: "curves overlay" could be rephrased for clarity, e.g., "their coverage increase curves overlaid".

L277: "KEEG-Decoder" should likely be "KEGG-Decoder". This is repeated on line 281.

L213-223 This analysis is not yet concerning the de-novo assembly but rather still the mapping-based coverage and thus could be part of the previous paragraph. Similarly, the panels Fig 2 A, B could be part of Fig 1 or removed as they are only a different visualisation of a part of the heatmap in Fig 1A.

Fig 2C the title Mock-log should probably be mock-stag
L284 same log vs. stag

Reviewer #3

(Remarks to the Author)

This is an excellent research study done on somewhat complex mock communities, aimed at studying the interplay between sequencing coverage and strain variation on taxonomic, MAG, and functional analysis of microbial communities.

The major results are that taxonomic analysis of these communities should be accurate, given the coverage and recovery of the known source genomes in the metagenomic sequencing; that strain-resolved MAG analysis is challenging, at best, and baldly inaccurate, at worst; and that functional analysis is challenging at low coverage.

The conclusions are well supported by the data and the analysis, and overall this is an excellent study that cleanly and substantially expands the literature in this important area.

To the extent that I have any critiques, they are mostly about the presentation of the work in the context of the larger field.

First, the strain complexity of these artificial communities should be compared to the expected strain complexity of real communities. My sense is that ~70 strains is reasonably comparable to the expected strain complexity of the dominant species in gut microbiota, but I would like the authors to make a statement here.

Second, the authors do not use a taxonomic workflow directly, but rather estimate the degree to which the known genomes are sequenced in the mixture. This is not an uninteresting question, but it only gets at one aspect of the situation: is the information there? A different question is: do taxonomic profilers get distracted by the reference database? Ideally the authors could use a system like Kraken or sourmash (which, in my expert opinion, should be able to recover the strains!) to ask how well a straightforward taxonomy tool actually works. I do recognize that this is in itself a lot of extra work so I would be quite satisfied if the authors simply noted that actual calling of taxonomy on this community might not work that well, given other considerations such as the reference database.

Signed, C. Titus Brown.

Decision Letter:

28th May 2025

Dear Professor Clavel,

Thank you for your patience while your manuscript "Benchmarking of shotgun sequencing depth highlights strain-level limitations of metagenomic analysis" was under peer review at Nature Microbiology. It has now been seen by our referees, whose expertise and comments you will find at the end of this email. In the light of their advice, we have decided that we cannot offer to publish your manuscript in Nature Microbiology.

From the reports, you will see that while they find your work of some potential interest, the referees raise concerns regarding the robustness of the analysis. They have highlighted technical issues regarding sequence mapping, binning, small size of the dataset, and pooling of samples with limited replication among other technical issues. They have also mentioned that although it addresses an important issue, the work needs to provide a clearer direction on solving those issues. Unfortunately, these criticisms are sufficiently important as to preclude publication of your work in Nature Microbiology.

Although we cannot offer to publish your manuscript, I have discussed your manuscript and the reviewers' comments with our colleague Javier Martinez Vesga (javier.martinez-vesga@nature.com) at Nature Communications. He would be happy to send a suitably revised version back to the reviewers -when available-.

Should you wish to have a revised paper re-reviewed at Nature Communications, please use this link: Link Redacted>manuscript transfer portal.

Please provide a detailed point-by-point to all the reviewer's comments and a Cover Letter summarizing the changes.

If there is anything you would like to discuss before transferring the paper and its reviews, please don't hesitate to contact Javier by e-mail.

Please note that Nature Communications is a fully open access journal. For information about article processing charges, open access funding, and advice and support from Springer Nature, please consult the Nature Communications Open Access page (www.nature.com/ncomms/open_access/index.html).

Should you wish to have a revised paper re-reviewed at Nature Communications, please use this link: Link Redacted>manuscript transfer portal.

Please provide a detailed point-by-point to all the reviewer's comments and a Cover Letter summarizing the changes.

If there is anything you would like to discuss before transferring the paper and its reviews, please don't hesitate to contact Javier by e-mail.

Please note that Nature Communications is a fully open access journal. For information about article processing charges, open access funding, and advice and support from Springer Nature, please consult the Nature Communications Open Access page (www.nature.com/ncomms/open_access/index.html).

rs/author_resources/transfer_manuscripts.html?WT.mc_id=EMI_NPG_1511_AUTHORTRANSF&WT.ec_id=AUTHOR>manuscript transfer FAQ page.

I am sorry that we cannot be more positive on this occasion, but hope that you find the referees' comments helpful when preparing your paper for resubmission elsewhere.

Yours sincerely,

Reviewer Expertise:

Referee #1: Environmental metagenomics

Referee #2: Metagenomics

Referee #3: Omics methods development

Reviewers Comments:

Reviewer #1 (Remarks to the Author):

This manuscript addresses some very important questions about the accuracy, reproducibility and interpretability of metagenome datasets. Many metagenomic studies generate, analyze and interpret metagenome data without any significant effort to validate or ground-truth the methods use, and only a small number of studies have aimed to benchmark metagenomic workflows against a "gold standard". This is challenging to do, since the true nature of "real" communities is unknown and synthetic communities fail to reproduce the complexity of natural communities, particularly in terms of strain heterogeneity. Synthetic data, also often used for this type of study, fails to reproduce the potential biases resulting from the library construction and sequencing process. Some of these weaknesses are addressed in the current study.

Although the authors introduce the study with a focus on the utility of shallow metagenome sequencing, and the depth necessary to gain phylogenetic and functional insights, their most notable findings relate to deep metagenomics – in particular the sometimes very poor quality of "high quality" metagenome assembled genomes. Overall, they find that shallow sequencing is effective for phylogenetic and functional characterization only when a high-quality reference database is available, but that such a high-quality database cannot be generated based on MAGs from deep sequencing. This is based on the finding that MAGs, even those derived from deep sequencing that appear high quality based on standard metrics, are frequently chimeric and/or redundant and do not accurately represent the functional gene complement of the source genomes.

While this finding is notable, it is unfortunately based on a relatively limited data set that is not fully explored. While the authors stress the weaknesses of other benchmarking studies based on deep sequencing of native communities, bioinformatic subsampling of data and/or in silico generation of mock communities, the specific details of their protocols for generating datasets of different depths are glossed over in multiple places. It appears that different libraries were constructed for each target sequencing depth, and then "pooled to reflect the nine different sequencing depths targeted" – it would have been helpful to see more about how this was done, since the results are presented as very precise basecounts per library (e.g. 0.1 Gb, 0.25 Gb, etc.) – did the pooling strategy really yield exactly the sequence amounts desired, or was there some bioinformatic subsampling? More importantly, was there really only one library made per target depth with no replication? For a benchmarking study, wouldn't proper replication and statistics be critical to any findings? The resultant lack of statistical significance for any of the findings makes essentially all of the observations anecdotal.

Another key weakness is the use of just one sequence assembly method, MegaHIT. While a comprehensive analysis of different assemblers would be a major effort in itself, no justification is given for choosing this particular assembler over many others that could potentially produce higher quality and less chimeric MAGs. Without further investigation of this question, the assertion that MAGs fail to accurately represent the source genomes is unwarranted.

I also thought the authors downplayed the contributions of previous studies, such as Bowers et al 2015 (reference 43) which also made libraries from mixed isolate DNA for metagenome sequencing but was only briefly mentioned in the discussion. There is also a large metagenome benchmarking effort called the Critical Assessment of Metagenome Interpretation (CAMI; <https://camii-challenge.org/>) which is more focused on bioinformatics tools but addresses similar questions and does not appear to be referenced at all.

Overall, I appreciate the effort to ground truth common metagenomic approaches, and the recognition that investigators may put too much faith in the tools they use. But this limited study only provides a glimpse at how things can go wrong and doesn't offer any novel solutions, or guidance that extends much beyond the specific communities examined.

A few more specific comments on the manuscript are below.

Line 43: "Shallow metagenomics, defined as shotgun sequencing at a depth ≤ 1 Gb (ca. 3 million reads)..." Where does this definition come from?

Line 94: "Fragment size of cleaned DNA was determined on an Agilent D1000 Tapestation..." What was the fragment size of these libraries?

Line 288: I'm not sure an equation is needed for this simple arithmetic. However, if the insert size is 250 bp, doesn't this equation overestimate the amount of unique sequence generated by treating each 150 bp read as representing unique sequence?

Figure 2A: The colors for strain CLA-AA-H239 and CR B11 are nearly identical.

Line 361: "However, sequencing depths >5 Gb are required to obtain nearly full genome coverage..." This seems wholly dependent on the composition of the community. A similar statement is made on line 382 ("...showing that >2 Gb is required to reach $>50\%$ completeness.")

Lines 374-380: This section suggests potential problems with binning and ways to improve, but glaringly ignores potential assembly errors.

Lines 44, 79, 273: There should be no comma after "both"

Line 281: "KEEG" should be "KEGG"

Reviewer #2 (Remarks to the Author):

The manuscript by Treichel NS, et al., presents the impact of sequencing depth on metagenomic analysis. The authors used two mock bacterial DNA communities (even and staggered distributions) and analyzed them at nine different sequencing depths (from 0.1 Gb to 10.0 Gb). The study also explored the influence of different library preparation protocols and the presence of host DNA on the results. The key findings highlight that while reference-based taxonomic and strain-level insights can be retrieved at relatively shallow sequencing depths (0.5-1.0 Gb), de novo metagenome-assembled genome (MAG) reconstruction requires significantly deeper sequencing (>10 Gb) and often results in chimeric MAGs. Functional pathway analysis was found to be reliable at 2 Gb, but comprehensive proteome coverage required 10 Gb of sequencing data.

Major points

The study evaluates the impact of actual variations in sequencing depth, which is highly valuable for practical experimental design. As a complementary perspective, the authors might consider discussing or, if feasible, briefly exploring the use of technical subsampling (rarefaction). By taking the deepest sequenced libraries and computationally subsampling reads to simulate shallower depths, it could help further disentangle limitations arising purely from statistical effects of reduced read counts versus potential biases introduced during library preparation or the sequencing process itself when targeting lower depths from the outset. This could add another dimension to the interpretation of why certain analytical goals are challenging at shallower depths.

Staggered community: What is the distribution? (e.g., log-normal? Or others?)

Line 56-58, downsampling sequencing of higher depth vs. directly sequencing at lower depth – citation or comparison here?

L176-182, the authors used coverM with bwa-mem mapper. Since the reference genomes are sometimes very similar (different strains of the same species), are multi-mapping reads showing up, or is every read uniquely assigned to the correct reference? If multi-mapping reads exist, how are these handled, and how does it affect the coverage calculation? Can it explain cases with very low coverage, and that almost all genomes never achieve 100% coverage at any depth? Also related to analysis at L217-223, for genome pairs with higher ANI, multi-mapping reads could be more likely.

L200-204 Are the outlier relative abundances due to imprecision in the lab or due to some computational artifact?

L234 "This suggests MAGs become increasingly fragmented with greater sequencing depth, rather than coalescing into a single high-quality MAG". This statement doesn't immediately follow from the data in Fig. 2C, as both the number of MAGs and the number of references covered are increasing. The fragmentation appeared not to be assessed. A more in-depth discussion or hypothesis as to why this occurs would also strengthen the paper. Could this be related to the assembly algorithms used or inherent complexities in resolving highly similar genomes?

L244-249. Are the different contigs of the final MAG coming from different strains? According to the Supp. Table 3, they were assigned to multiple ref. genomes. Is it possible that the contigs themselves were assembled from reads coming from multiple strains? There is a whole class of assembly and binning tools that attempt "strain-aware" metagenomic assembly, while the tools used by the authors don't consider such cases, and so a single bin per species, possibly a strain hybrid/chimera, is expected.

Maybe the authors could at least comment on this in the manuscript.

L250-263, Fig2D. The authors use checkM to estimate the completeness and contamination of the MAGs, but having the ground truth genomes available and having done the mapping, the authors could provide more exact values of true completeness (coverage of the assigned reference) and contamination (coverage of other references). The distinction of whether the MAG was assigned to one or multiple references is not very meaningful, as this doesn't reflect the size of the portion of the MAG coming from other references – the contamination in the checkM sense of meaning. I understand it's not straightforward to visualize in one plot, but having shown how with increasing seq depth the number of MAGs with different true completeness and true contamination changes would be more illustrative.

Moreover, there is a drop in the number of MAGs from 5Gb to 10Gb in the staggered case, which could support the previous statement on L334 - with higher depth, the tool tends to over-bin the contigs. Has this been reported previously in the literature? Is this specific to the binner used by the authors? As a simpler analysis preceding the MAGs, the authors could consider working just with contigs without binning and assessing how much the reference genomes are covered by them at different depths, as the chimeric MAGs are a limitation of the binner.

L265-265 "de-novo analysis based on MAG reconstruction leads to contaminated genomes, even at supposed high-quality". The high-quality definition allows for contamination up to 5%, the authors could show how much of the true contamination there is based on the coverage of ref. genomes, but it would be just performing a benchmark of checkM prediction accuracy. Instead, the authors could investigate whether higher sequencing depth and thus higher coverage of certain genomes will lead to lower contamination or not.

L266-268 "Together with the creation of multiple MAGs for individual strains, these findings suggest that strain-level diversity within MAG catalogues is artificially inflated". Does it happen that one ref. genome would be represented by multiple MAGs that are of medium or high completeness, and thus they overlap? Or is it that their contigs get split into multiple disjoint sets? To support the statement of the strain-level diversity being inflated, the authors would have to show that a single strain could be assembled and binned to multiple MAGs that overlap each other, and moreover they would pass certain de-replication thresholds used by MAG catalogues. I believe this analysis would be of interest to the readers.

L342-363 The authors should highlight the limitation of their analysis, that is, that they always mapped the reads against the reference genomes of the exact strains they were looking for. This might be similar to some applications of pathogen detection where the exact strain is known, but for general microbiome analyses, usually only the ref. genomes of other strains of the species are available, and genomes of all possible species are included in the database. This might reduce both the sensitivity and also introduce false positives, leading to a lower signal-to-noise ratio in the lower seq depths than the ideal case presented here. It should also be mentioned that k-mer or marker gene-based approaches will have different merits and limitations compared to the full genome mapping considered here.

The authors show that library preparation and presence of background DNA significantly impact the results, especially at lower seq depths. It would be nice to discuss more about the influence of these factors, for example, does lower template DNA or more PCR cycles specifically lead to biases or reduced coverage? The manuscript mentions that Facility 1, using more template DNA and fewer PCR cycles, yielded more robust profiles. It would be valuable to discuss whether differences between facilities could also be due to other factors not taken into account (e.g., bead cleanup?).

Minor points

The title should include "shallow sequencing" since that is the target application of the benchmark

L61: "Given the urgent need to better define the strengths and limitations of shallow metagenomics, we systematically evaluated..." This sentence appears twice consecutively.

L111: "biopython13" should likely be "Biopython".

L185-186: "Thomasclavelia ramosa CLA-JM-H52 and Enterocloster clostridioformis CLA-JM-H51, which were added to the Mock with highest DNA concentration (17.2%; 40 ng), and V. intestinalis CLA-AV-13, added at a low concentration (0.86%, 2 ng)." Ensure clarity that V. intestinalis reaching 100% coverage despite low input is a notable point, perhaps linked to its genome characteristics or sequencing efficiency.

L221: "curves overlay" could be rephrased for clarity, e.g., "their coverage increase curves overlaid".

L277: "KEEG-Decoder" should likely be "KEGG-Decoder". This is repeated on line 281.

L213-223 This analysis is not yet concerning the de-novo assembly but rather still the mapping-based coverage and thus could be part of the previous paragraph. Similarly, the panels Fig 2 A, B could be part of Fig 1 or removed as they are only a different visualisation of a part of the heatmap in Fig 1A.

Fig 2C the title Mock-log should probably be mock-stag
L284 same log vs. stag

Reviewer #3 (Remarks to the Author):

This is an excellent research study done on somewhat complex mock communities, aimed at studying the interplay between sequencing coverage and strain variation on taxonomic, MAG, and functional analysis of microbial communities.

The major results are that taxonomic analysis of these communities should be accurate, given the coverage and recovery of the known source genomes in the metagenomic sequencing; that strain-resolved MAG analysis is challenging, at best, and baldly inaccurate, at worst; and that functional analysis is challenging at low coverage.

The conclusions are well supported by the data and the analysis, and overall this is an excellent study that cleanly and substantially expands the literature in this important area.

To the extent that I have any critiques, they are mostly about the presentation of the work in the context of the larger field.

First, the strain complexity of these artificial communities should be compared to the expected strain complexity of real communities. My sense is that ~70 strains is reasonably comparable to the expected strain complexity of the dominant species in gut microbiota, but I would like the authors to make a statement here.

Second, the authors do not use a taxonomic workflow directly, but rather estimate the degree to which the known genomes are sequenced in the mixture. This is not an uninteresting question, but it only gets at one aspect of the situation: is the information there? A different question is: do taxonomic profilers get distracted by the reference database? Ideally the authors could use a system like Kraken or sourmash (which, in my expert opinion, should be able to recover the strains!) to ask how well a straightforward taxonomy tool actually works. I do recognize that this is in itself a lot of extra work so I would be quite satisfied if the authors simply noted that actual calling of taxonomy on this community might not work that well, given other considerations such as the reference database.

Signed, C. Titus Brown.

***Nature Communications* is the Nature Portfolio flagship Open Access journal. If you would like this work to be considered for publication there, you can easily transfer the manuscript by following the instructions below. It is not necessary to reformat your paper. Once all files are received, the editors at *Nature Communications* will assess your manuscript's suitability for potential publication; they aim to provide feedback quickly, with a median decision time of 8 days for first editorial decisions on suitability. Since your paper has been peer reviewed at this journal, the referee reports will also be transferred and assessed by the editorial team. In some cases, papers are accepted without further peer review, providing a rapid path to publication. The journal is also proud to offer double blind and transparent peer review options. For 2021, the 2-year impact factor for *Nature Communications* is 17.694 and the 2-year median is 10 (for further information on journal impact factors, please visit our http://www.nature.com/npg_company_info/journal_metrics.html>Nature journals metrics page). Our http://www.nature.com/ncomms/open_access/index.html>open access pages contain information about article processing charges, open access funding, and advice and support from Springer Nature.

** Although we cannot offer to publish your manuscript, we believe the editors at our sister journal, *Communications Biology*, will find it interesting and recommend you transfer it there. *Communications Biology* is a selective Nature Portfolio title publishing Open Access research that brings new insight in all areas of the biological sciences. <https://www.nature.com/commsbio/journal-information/journal-impact>>Additional journal metrics and information can be found here. Their editors prioritise good author service, fast peer review (in 2021, the median time to decision after first review was 40 days), and are happy to answer any questions you may have <mailto:commsbio@nature.com>>(commsbio@nature.com). The journal has an Impact Factor of 6.548, a CiteScore of 6.0 and a Scimago quartile ranking of Q1.

Please note that *Communications Biology* is a fully open-access journal and an article processing charge will apply to any papers accepted for publication. Our <https://www.nature.com/commsbio/about/open-access>>open access pages contain information about article processing charges, open access funding, and advice and support from Springer Nature.

If you wish to transfer your manuscript to *Communications Biology*, please use our manuscript transfer portal using the link

below to initiate the transfer to this journal (or to another journal of your choice in the Nature Research portfolio). If you transfer to Nature-branded journals or to the Communications journals, you will not have to re-supply manuscript metadata and files. This link can only be used once and remains active until used. For more information, please see our [manuscript transfer FAQ](https://www.nature.com/nature-portfolio/for-authors/transfer) page.

****Scientific Reports** publishes primary research from all areas of the natural and clinical sciences that is judged to be scientifically valid and technically sound, whatever the considered significance. If you would like this work to be considered for publication in *Scientific Reports*, you can easily transfer the manuscript by following the instructions below. Your manuscript will be handled by an academic scientist who is an Editorial Board Member and will manage the peer review process and decide whether a paper should be accepted for publication. Most submissions are peer reviewed by one or more referees as well as the editorial board member and you can expect to receive an editorial decision within 56 days. Over 55% of the papers are published following peer review.

To discover more about this journal and, should you wish, have your paper considered the Editorial Board of *Scientific Reports*, please use the link to the manuscript transfer service provided in the footnote below. Please see our [open access pages](http://www.nature.com/ncomms/open_access/index.html) for information about article processing charges, open access funding, and advice and support from Springer Nature.

Version 2:

Decision Letter:

5th January 2026

Dear Professor Clavel,

Thank you for your letter asking us to reconsider our decision on your Article entitled "Benchmarking of shotgun sequencing depth highlights limitations of strain-level analysis and shallow metagenomics". After careful consideration we have decided that we would be willing to consider a revised version of your manuscript.

Along with your revised manuscript, you should also submit a separate point-by-point response to all of the concerns raised by the referees, in each case describing what changes have been made to the manuscript or, alternatively, if no action has been taken, providing a compelling argument for why that is the case. If we feel that a substantial attempt has been made to address the referees' comments, this response will be sent back to the referees - along with the revised manuscript - so that they can judge whether their concerns have been addressed satisfactorily or otherwise.

- ensure it complies with our format requirements for Letters as set out in our guide to authors at www.nature.com/nmicrobiol/authors/index.html

- state in a cover note the length of the text, methods and legends; the number of references and the number of display items.

Please ensure that all correspondence is marked with your Nature Microbiology reference number in the subject line.

Please use the following link to submit your revised manuscript:

Link Redacted

We hope to receive your revised paper within four weeks. If you cannot send it within this time, please let us know so that we can close your file. In this event, we will still be happy to reconsider your paper at a later date so long as nothing similar has been accepted for publication at Nature Microbiology or published elsewhere in the meantime. Should you miss the four-week deadline and your paper is eventually published, the received date will be that of the revised, not the original, version.

I would appreciate it if you could tell me if you think you will be able to submit a revised manuscript, and also the likely timescale.

I look forward to hearing from you soon.

Yours sincerely,

Version 3:

Reviewer comments:

Reviewer #1

(Remarks to the Author)

I appreciate the considerable work that has gone into improving the manuscript, in particular the additional data generated for

confirmation and statistical power as well as the additional analyses performed. I also appreciate the thought put into assessing the limitations and offering useful suggestions for microbiome researchers. The new version is much improved but I have a few more questions and suggestions noted below.

Abstract line 29: "...even high-quality MAGs were chimeric, with 54.5 to 81.8% accurately representing the original strains..." Maybe "high-quality" should be in quotes here? Or else say "even MAGs deemed high quality by standard metrics"?
Abstract line 32: "Functionally, 2 Gb provided reliable insights at the pathway level..." I still disagree with asserting that a certain amount of sequence is sufficient without noting that this is only known to be true for the communities examined, e.g. "Functionally, 2 Gb provided reliable insights at the pathway level for each of the communities tested..."
Line 41: "...at an ever-increasing sample size." Do you mean sample number?
Line 251: "For Mock-even-70, relative abundances oscillated around the expected value..." I think "oscillated" is a poor choice here, something like "clustered" would be better.

Response to reviews:

On response 10 about the depth of sequencing – it's true that, for example, 333,333 paired-end 150 bp read pairs will generate 0.1 Gb of raw sequence. However, if those paired-end 150 bp reads are generated on 250 bp inserts, they cannot generate more than 250 bp of unique sequence even if 300 high-quality bases are produced. It's not a critical point, but the coverage obtained with 0.1 Gb raw sequence would be different if, for example, you were to sample 666,666 single-ended reads from the dataset.

(Remarks on code availability)

Reviewer #2

(Remarks to the Author)

The revised manuscript has improved significantly, and this reviewer thanks the authors for their effort in updating their analyses.

(Remarks on code availability)

Reviewer #3

(Remarks to the Author)

I appreciate the substantial revisions and have no further comments. Thank you!

(Remarks on code availability)

Decision Letter:

Our ref: NMICROBIOL-25010326C

9th February 2026

Dear Dr. Clavel,

Thank you for submitting your revised manuscript "Benchmarking of shotgun sequencing depth highlights limitations of strain-level analysis and shallow metagenomics" (NMICROBIOL-25010326C). It has now been seen by the original referees and their comments are below. The reviewers find that the paper has improved in revision, and therefore we'll be happy in principle to publish it in Nature Microbiology, pending minor revisions to satisfy the referees' final requests and to comply with our editorial and formatting guidelines.

Thank you again for your interest in Nature Microbiology Please do not hesitate to contact me if you have any questions.

Sincerely,

Reviewer #1 (Remarks to the Author):

I appreciate the considerable work that has gone into improving the manuscript, in particular the additional data generated for confirmation and statistical power as well as the additional analyses performed. I also appreciate the thought put into assessing

the limitations and offering useful suggestions for microbiome researchers. The new version is much improved but I have a few more questions and suggestions noted below.

Abstract line 29: "...even high-quality MAGs were chimeric, with 54.5 to 81.8% accurately representing the original strains..."

Maybe "high-quality" should be in quotes here? Or else say "even MAGs deemed high quality by standard metrics"?

Abstract line 32: "Functionally, 2 Gb provided reliable insights at the pathway level..." I still disagree with asserting that a certain amount of sequence is sufficient without noting that this is only known to be true for the communities examined, e.g.

"Functionally, 2 Gb provided reliable insights at the pathway level for each of the communities tested..."

Line 41: "...at an ever-increasing sample size." Do you mean sample number?

Line 251: "For Mock-even-70, relative abundances oscillated around the expected value..." I think "oscillated" is a poor choice here, something like "clustered" would be better.

Response to reviews:

On response 10 about the depth of sequencing – it's true that, for example, 333,333 paired-end 150 bp read pairs will generate 0.1 Gb of raw sequence. However, if those paired-end 150 bp reads are generated on 250 bp inserts, they cannot generate more than 250 bp of unique sequence even if 300 high-quality bases are produced. It's not a critical point, but the coverage obtained with 0.1 Gb raw sequence would be different if, for example, you were to sample 666,666 single-ended reads from the dataset.

Reviewer #2 (Remarks to the Author):

The revised manuscript has improved significantly, and this reviewer thanks the authors for their effort in updating their analyses.

Reviewer #3 (Remarks to the Author):

I appreciate the substantial revisions and have no further comments. Thank you!

Version 4:

Decision Letter:

18th March 2026

Dear Professor Clavel,

I am pleased to accept your Article "Benchmarking of shotgun sequencing depth reveals the potential and limitations of shallow metagenomics and strain-level analysis" for publication in Nature Microbiology. Thank you for having chosen to submit your work to us and many congratulations.

You may wish to make your media relations office aware of your accepted publication, in case they consider it appropriate to organize some internal or external publicity. Once your paper has been scheduled you will receive an email confirming the publication details. This is normally 3-4 working days in advance of publication. If you need additional notice of the date and time of publication, please let the production team know when you receive the proof of your article to ensure there is sufficient time to coordinate. Further information on our embargo policies can be found here:

<https://www.nature.com/authors/policies/embargo.html>

Authors may need to take specific actions to achieve compliance with funder and institutional open access mandates. If your research is supported by a funder that requires immediate open access (e.g. according to [Plan S principles](https://www.springernature.com/gp/open-science/plan-s-compliance) or the [NIH public access policy](https://www.springernature.com/gp/open-science/us-federal-agency-compliance)) then you should select the gold OA route, and we will direct you to the compliant route where possible. Because authors warrant under our subscription licensing terms that they haven't committed to licensing any version of their article under a licence inconsistent with the terms of our agreement – including the applicable embargo period – publication under the subscription model isn't suitable for authors whose funders require no embargo.

With kind regards,

P.S. Click on the following link if you would like to recommend Nature Microbiology to your librarian
<http://www.nature.com/subscriptions/recommend.html#forms>

** Visit the Springer Nature Editorial and Publishing website at http://editorial-jobs.springernature.com?utm_source=ejP_NMicro_email&utm_medium=ejP_NMicro_email&utm_campaign=ejp_NMicro for more information about our career opportunities. If you have any questions please click [here](mailto:editorial.publishing.jobs@springernature.com).

Reviewer #1 (Remarks to the Author):

Referee #1: Environmental metagenomics

- 1- This manuscript addresses some very important questions about the accuracy, reproducibility and interpretability of metagenome datasets. Many metagenomic studies generate, analyze and interpret metagenome data without any significant effort to validate or ground-truth the methods use, and only a small number of studies have aimed to benchmark metagenomic workflows against a “gold standard”. This is challenging to do, since the true nature of “real” communities is unknown and synthetic communities fail to reproduce the complexity of natural communities, particularly in terms of strain heterogeneity. Synthetic data, also often used for this type of study, fails to reproduce the potential biases resulting from the library construction and sequencing process. Some of these weaknesses are addressed in the current study.

Answer: Thank you for recognizing the importance of using reference datasets to validate metagenomic methods commonly used by many.

In the revised version, we have increased the number and complexity of the mock communities further: we added (i) 22 newly sequenced datasets, including long-read sequencing, and (ii) 10 random subsampling of a high-complexity mock (70 members, staggered distribution) for each sequencing depth followed by statistical analyses. Whilst we agree with the reviewer that the mock communities used are still simplifications of native communities (e.g. include only bacteria, as now acknowledged in the new ‘Limitations’ section within the discussion, L511-528), working with known mixtures is essential for benchmarking, and their diversity surpasses precedent in the literature. Combined with the addition of background (non-target) DNA and real sequencing of different libraries for each sequencing depth, the work is quite extensive and considers potential experimental bias.

- 2- Although the authors introduce the study with a focus on the utility of shallow metagenome sequencing, and the depth necessary to gain phylogenetic and functional insights, their most notable findings relate to deep metagenomics – in particular the sometimes very poor quality of “high quality” metagenome assembled genomes. Overall, they find that shallow sequencing is effective for phylogenetic and functional characterization only when a high-quality reference database is available, but that such a high-quality database cannot be generated based on MAGs from deep sequencing. This is based on the finding that MAGs, even those derived from deep sequencing that appear high quality based on standard metrics, are frequently chimeric and/or redundant and do not accurately represent the functional gene complement of the source genomes.

While this finding is notable, it is unfortunately based on a relatively limited data set that is not fully explored.

Answer: Following this and the following comments, we have substantially expanded the scope of our study and the mock communities used. Based on real sequencing of separate libraries at different sequencing depths, which we and the reviewers agreed is a strength, and now multiple new additions (more mock communities, third sequencing facility, long-read sequencing), the revised version of the manuscript contains 94 datasets, which we think is comprehensive. With this strategy, the work includes two important layers of confirmation:

1) Separate sequencing of three communities at all sequencing depths, enabling the validation of readouts with independent datasets. This includes sequencing in different facilities to confirm the results.

2) Statistical comparison of replicates from subsampling ($N = 10$ per sequencing depth), as also suggested by Reviewer 2. This new data confirmed our previous findings, and it also helped us working towards solutions regarding chimeric MAGs (see our answer to comment 7 below).

3- While the authors stress the weaknesses of other benchmarking studies based on deep sequencing of native communities, bioinformatic subsampling of data and/or in silico generation of mock communities, the specific details of their protocols for generating datasets of different depths are glossed over in multiple places. It appears that different libraries were constructed for each target sequencing depth, and then “pooled to reflect the nine different sequencing depths targeted” – it would have been helpful to see more about how this was done, since the results are presented as very precise basecounts per library (e.g. 0.1 Gb, 0.25 Gb, etc.) – did the pooling strategy really yield exactly the sequence amounts desired, or was there some bioinformatic subsampling?

Answer: We apologize for not making this clearer in the original text. We confirm that separate libraries were prepared and sequenced to replicate as closely as possible the variability of a sequencing project with real samples. The libraries were pooled for multiplexed sequencing, but their specific indices enabled separate data analysis. As the reviewer correctly pointed out, precise sequence numbers cannot be obtained with this empirical strategy. However, to enable precise benchmarking and since it is easier for readers to understand data when exact sequencing depths are used, minor subsampling was performed on each dataset (i.e., per library) to obtain the target sequencing depth (0.10, 0.25, 0.50, 0.75, 1.0, 1.5, 2.5, 5.0, 10.0, 20.0, or 50.0 Gb). For the sake of clarity, the exact number of sequences obtained after sequencing (NovaSeq6000 PE 150 bp) per forward or reverse file are now stated in New Supplementary Table S3 with the respective number of subsampled reads.

4- More importantly, was there really only one library made per target depth with no replication? For a benchmarking study, wouldn't proper replication and statistics be critical to any findings? The resultant lack of statistical significance for any of the findings makes essentially all of the observations anecdotal.

Answer: After adding several new datasets during the revisions, the new version of the manuscript includes two layers of replication:

- (i) Separate sequencing of three main mock communities (Mock-even-70, Mock-stag-24, new Mock-stag-70), showing consistency of the readouts across independent datasets.
- (ii) Subsampling with multiple replicates ($N = 10$ for each sequencing depth), followed by statistical testing.

In addition, our detailed investigations of chimeric MAGs now include two additional sequencing depths (20 and 50 Gb), long-read sequencing in a third facility, and 10 new versions of Mock-stag-24 differing in their composition (Mock-stag-24 v1-10), which represent additional layers of validation of the previous results. With of all this, the main findings presented in the original manuscript regarding taxonomic and functional coverage and the accuracy of MAGs are confirmed. Accordingly, Figs. 1-3 were modified extensively.

- 5- Another key weakness is the use of just one sequence assembly method, MEGAHIT. While a comprehensive analysis of different assemblers would be a major effort in itself, no justification is given for choosing this particular assembler over many others that could potentially produce higher quality and less chimeric MAGs. Without further investigation of this question, the assertion that MAGs fail to accurately represent the source genomes is unwarranted.

Answer: The original procedure included MEGAHIT because it is a standard approach used by many. We have addressed the comment by comparing this original procedure with another commonly used approach, metaSPAdes. The latter performed a little better in terms of chimeric MAGs. The data is presented in new Fig. 2E and 2F and corresponding text.

- 6- I also thought the authors downplayed the contributions of previous studies, such as Bowers et al 2015 (reference 43) which also made libraries from mixed isolate DNA for metagenome sequencing but was only briefly mentioned in the discussion. There is also a large metagenome benchmarking effort called the Critical Assessment of Metagenome Interpretation (CAMI; <https://cami-challenge.org/>) which is more focused on bioinformatics tools but addresses similar questions and does not appear to be referenced at all.

Answer: It was not our intention to discredit the work of others, and we apologize if this was perceived as such. The data obtained by Bowers et al. (2015) is now clearly referenced in the revised discussion when stressing the limitations of our study. We are aware of CAMI and agreed that it should be quoted; thank you for pointing this out. We did not do so originally because the published version of CAMI mainly focused on tool comparisons. We now refer to CAMI at the end of the manuscript when mentioning that future work will facilitate further progress at a larger scale (beyond what can be achieved by a single group, as in this work).

- 7- Overall, I appreciate the effort to ground truth common metagenomic approaches, and the recognition that investigators may put too much faith in the tools they use. But this limited study only provides a glimpse at how things can go wrong and doesn't offer any novel solutions, or guidance that extends much beyond the specific communities examined.

Answer: Thank you for mentioning the relevance of our study.

Regarding the statement that the study is limited: We have commented above on the many new experiments and analyses done to enhance the scale of this work further and thereby consolidate the findings, which strengthened the main conclusions. The work includes multiple layers of replication (independent datasets and replicates with statistical analysis), which confirmed the original results.

Regarding the need to work on solutions: It was a good suggestion, thank you. The revised version includes two assembly tools, cross-sample binning, and long-read sequencing as potential ways to improve the issue with chimeric MAGs. Hence, we have also worked towards providing some solutions.

In conclusion, we agree with the reviewer that it is essential to make investigators aware of the limitations of the methods they are using, and we feel this manuscript achieves this by presenting new information and comprehensive data. To enhance transparency, we have now added a new section to the end of the discussion that clearly presents the limitations of our own work, including the nature of the reference communities we have worked with (L511-528).

A few more specific comments on the manuscript are below.

- 8- Line 43: "Shallow metagenomics, defined as shotgun sequencing at a depth ≤ 1 Gb (ca. 3 million reads)..." Where does this definition come from?

Answer: There is no clear definition of shallow metagenomics we are aware of, as this is a rather empirical concept. The sentence is based on a consensus from published work. We have rephrased the sentence as follows: "Shallow metagenomics, commonly referred to as shotgun sequencing at a depth ≤ 1 Gb (ca. 3 million reads)..." Previous articles that have tested shallow metagenomics are quoted at the end of this sentence.

- 9- Line 94: "Fragment size of cleaned DNA was determined on an Agilent D1000 TapeStation..." What was the fragment size of these libraries?

Answer: Thank you. The information (~320 bp library fragment size \triangleq ~200 bp genomic DNA insert size) was added to the sentence.

- 10- Line 288: I'm not sure an equation is needed for this simple arithmetic. However, if the insert size is 250 bp, doesn't this equation overestimate the amount of unique sequence generated by treating each 150 bp read as representing unique sequence?

Answer: We based the calculations on the length of the raw reads directly after sequencing (150 bp), as the decision of how many samples should be pooled is usually based on the manufacturer's information of giga bases (Gb) per flow cell. The forward and reverse reads were merged during the workflow later, therefore the insert length was chosen to be larger than 150 bp.

Here is an example to illustrate our approach: 2,000,000,000 bp/150bp sequence length determined by NovaSeq kit = 13,333,333 reads of 150 bp length. As we used paired end mode: 13,333,333 reads/2 = 6,666,666 reads per R1 or R2. Therefore, the forward and reverse files for 2 Gb sequencing depth were subsampled to 6,666,666 reads.

Following this comment, and also Comment 3 (above), we have removed the equation from the text and now provide the exact number of sequences obtained after sequencing in New Supplementary Table S3 with the respective number of subsampled reads.

- 11- Figure 2A: The colors for strain CLA-AA-H239 and CR B11 are nearly identical.

Answer: Thank you for noticing this. The colors have been adjusted; Strain CLA-AA-H239 is now shown with a brighter color (new Suppl. Fig. S7).

- 12- Line 361: "However, sequencing depths > 5 Gb are required to obtain nearly full genome coverage..." This seems wholly dependent on the composition of the community. A similar statement is made on line 382 ("...showing that > 2 Gb is required to reach $> 50\%$ completeness.")

Answer: Thank you for pointing out this ambiguity. With this statement we wanted to emphasize that, below the given Gb, genome coverage or KEGG pathway completeness was relatively low, even under simplified conditions. Throughout the paper, we have now paid attention to specify that the provided

values are extrapolated from the conditions tested (L274) or “must be adjusted depending on the expected abundance and genome size of the targeted strains in complex samples” (L464).

13- Lines 374-380: This section suggests potential problems with binning and ways to improve, but glaringly ignores potential assembly errors.

Answer: Following this comment (and Comment 5 above) and another comment by Reviewer 2, we have now tested two commonly used assembly tools (MEGAHIT and metaSPAdes) and included the results in the revised manuscript (new Fig. 2E; Supplementary Figure S10).

14- Lines 44, 79, 273: There should be no comma after “both”

Answer: Thank you, corrected.

15- Line 281: “KEEG” should be “KEGG”

Answer: Thank you, corrected.

Reviewer #2 (Remarks to the Author):

Referee #2: Metagenomics

- 1- The manuscript by Treichel NS, et al., presents the impact of sequencing depth on metagenomic analysis. The authors used two mock bacterial DNA communities (even and staggered distributions) and analyzed them at nine different sequencing depths (from 0.1 Gb to 10.0 Gb). The study also explored the influence of different library preparation protocols and the presence of host DNA on the results. The key findings highlight that while reference-based taxonomic and strain-level insights can be retrieved at relatively shallow sequencing depths (0.5-1.0 Gb), de novo metagenome-assembled genome (MAG) reconstruction requires significantly deeper sequencing (>10 Gb) and often results in chimeric MAGs. Functional pathway analysis was found to be reliable at 2 Gb, but comprehensive proteome coverage required 10 Gb of sequencing data.

Answer: Thank you for the accurate reading, and for your clear understanding of the analyses and the main findings.

Major points

- 2- The study evaluates the impact of actual variations in sequencing depth, which is highly valuable for practical experimental design. As a complementary perspective, the authors might consider discussing or, if feasible, briefly exploring the use of technical subsampling (rarefaction). By taking the deepest sequenced libraries and computationally subsampling reads to simulate shallower depths, it could help further disentangle limitations arising purely from statistical effects of reduced read counts versus potential biases introduced during library preparation or the sequencing process itself when targeting lower depths from the outset. This could add another dimension to the interpretation of why certain analytical goals are challenging at shallower depths.

Answer: To address this comment and another one by Reviewer 1 about the robustness of the data, we have sequenced a new complex mock community (70 strains, staggered distribution) at 11 sequencing depths. We then performed random subsampling from the 50 GB data to obtain 10 different datasets within each of 10 sequencing depths (0.1 – 20 GB). The data is presented in new Fig. 1B. With this new analysis, we were able to confirm the original findings and demonstrate that they are representative, as the average coefficient of variation (CV) for each sequencing depth, which represents the variability of the readouts presented (e.g., taxonomic coverage), remained below 5%. The statistically significant increase in CV values at lower sequencing depths indicates enhanced variability with shallow metagenomics. This data was also used for strain-level analysis with MAGs (new Fig. 2F) and functional coverage (new Fig. 3C).

- 3- Staggered community: What is the distribution? (e.g., log-normal? Or others?)

Answer: Thank you; the exact compositions of all mock communities are now provided in Supplementary Table S1.

- 4- Line 56-58, downsampling sequencing of higher depth vs. directly sequencing at lower depth – citation or comparison here?

Answer: We are unsure about the meaning of this comment. This sentence in the introduction is meant to present alternative approaches to sequencing mock communities separately at different sequencing depths as proposed in our work: [Other common practices are to bioinformatically subsample deeply sequenced datasets to mimic different sequencing depths or to generate artificial Mock communities in-silico. However, these approaches do not reflect the impact of real sample processing and sequencing in the laboratory]. We have now referenced the study by Hillmann *et al.* (2018), Tremblay *et al.* (2022), and Fritz *et al.* (2019) to make clear that we talk about previous work. Regarding our own work, the revised version contains a direct comparison of separate sequencing Vs. downsampling from high sequencing depth.

5- L176-182, the authors used coverM with bwa-mem mapper. Since the reference genomes are sometimes very similar (different strains of the same species), are multi-mapping reads showing up, or is every read uniquely assigned to the correct reference? If multi-mapping reads exist, how are these handled, and how does it affect the coverage calculation? Can it explain cases with very low coverage, and that almost all genomes never achieve 100% coverage at any depth? Also related to analysis at L217-223, for genome pairs with higher ANI, multi-mapping reads could be more likely.

Answer: We can confirm that the reads were uniquely assigned. Therefore, multiple mapping did not occur with this approach, which should prevent certain genomes from acting as a sink for reads and confounding the analysis. We have added this information [...] no multiple read mapping] when providing the parameters of the coverM analysis in the methods (L204).

6- L200-204, Are the outlier relative abundances due to imprecision in the lab or due to some computational artifact?

Answer: This is an interesting question. Whilst we think that working with real sequencing data is a strength (and the reviewers apparently had the same opinion), it was not possible to distinguish the effects of the wet lab procedure from those of the bioinformatic analysis. We are confident we can exclude imprecisions during manual processing. Our best guess is that library preparation and sequencing introduce biases we cannot control. This point (inability to distinguish the individual effects of certain variables within our workflows) has been included in the new 'Limitations' section that we added at the end of the discussion (L511-528).

7- L234 "This suggests MAGs become increasingly fragmented with greater sequencing depth, rather than coalescing into a single high-quality MAG". This statement doesn't immediately follow from the data in Fig. 2C, as both the number of MAGs and the number of references covered are increasing. The fragmentation appeared not to be assessed. A more in-depth discussion or hypothesis as to why this occurs would also strengthen the paper. Could this be related to the assembly algorithms used or inherent complexities in resolving highly similar genomes?

Answer: We have now tried 2 widely used assembly approaches (MEGAHIT and metaSPAdes), but the main outcome was unchanged: whilst the total number of MAGs exceeded the expected number of strains, multiple MAGs were assigned to the same reference strain resulting in not all reference genomes being represented by a MAG. We realized that the term "fragmented" might have been confusing, so we rephrased the sentence as follows: "This suggests that multiple MAGs are created per

reference genome with greater sequencing depth, rather than coalescing into a single high-quality MAG”

- 8- L244-249. Are the different contigs of the final MAG coming from different strains? According to the Supp. Table 3, they were assigned to multiple ref. genomes. Is it possible that the contigs themselves were assembled from reads coming from multiple strains? There is a whole class of assembly and binning tools that attempt “strain-aware” metagenomic assembly, while the tools used by the authors don’t consider such cases, and so a single bin per species, possibly a strain hybrid/chimera, is expected. Maybe the authors could at least comment on this in the manuscript.

Answer: This was a good suggestion. As mentioned in the previous answer, we have now used an additional assembly tool (metaSPAdes) and compared the data. Some level of chimerism is already occurring during the assembly. The data is presented in new Suppl. Fig. S10 and the corresponding text. In the new 'Limitations' section at the end of the discussion (L511-528), we now also comment that additional tools, not tested in this study, may perform better. Nevertheless, the workflows in our study are widely used, highlighting the extent of the problem.

- 9- L250-263, Fig2D. The authors use checkM to estimate the completeness and contamination of the MAGs, but having the ground truth genomes available and having done the mapping, the authors could provide more exact values of true completeness (coverage of the assigned reference) and contamination (coverage of other references). The distinction of whether the MAG was assigned to one or multiple references is not very meaningful, as this doesn’t reflect the size of the portion of the MAG coming from other references – the contamination in the checkM sense of meaning. I understand it’s not straightforward to visualize in one plot, but having shown how with increasing seq depth the number of MAGs with different true completeness and true contamination changes would be more illustrative.

Answer: We used CheckM because this is by far the most widely used method to assess the quality of MAGs. We think that benchmarking CheckM, whilst interesting, is beyond the scope of this study.

- 10- Moreover, there is a drop in the number of MAGs from 5Gb to 10Gb in the staggered case, which could support the previous statement on L334 - with higher depth, the tool tends to over-bin the contigs. Has this been reported previously in the literature? Is this specific to the binner used by the authors? As a simpler analysis preceding the MAGs, the authors could consider working just with contigs without binning and assessing how much the reference genomes are covered by them at different depths, as the chimeric MAGs are a limitation of the binner.

Answer: The revised version contains additional datasets (e.g., a complex staggered mock with 70 species) which consolidated the main findings, but the drop in MAGs number between 5 and 10 GB was not observed (new Fig. 2E-G). Using the two assembly methods aforementioned, the contigs were analyzed by aligning them to the reference genomes with blastn (perc_identity, 97%; evalue, 1e-10; alignment length, >150 bp), revealing that the majority matched one single reference genomes but a certain degree of chimerism was already observed at this step. The data is presented in Suppl. Fig. S10.

11- L265-265 “de-novo analysis based on MAG reconstruction leads to contaminated genomes, even at supposed high-quality”. The high-quality definition allows for contamination up to 5%, the authors could show how much of the true contamination there is based on the coverage of ref. genomes, but it would be just performing a benchmark of checkM prediction accuracy. Instead, the authors could investigate whether higher sequencing depth and thus higher coverage of certain genomes will lead to lower contamination or not.

Answer: We agree with the reviewer that benchmarking checkM is out of scope. As suggested in this comment, we have looked at the association between sequencing depth and calculated contamination (checkM). The results show no significant association (figure below). Given the comprehensive aspect of the work, we feel this information is not essential and was not included in the manuscript.

12- L266-268 “Together with the creation of multiple MAGs for individual strains, these findings suggest that strain-level diversity within MAG catalogues is artificially inflated”. Does it happen that one ref. genome would be represented by multiple MAGs that are of medium or high completeness, and thus they overlap? Or is it that their contigs get split into multiple disjoint sets? To support the statement of the strain-level diversity being inflated, the authors would have to show that a single strain could be assembled and binned to multiple MAGs that overlap each other, and moreover they would pass certain de-replication thresholds used by MAG catalogues. I believe this analysis would be of interest to the readers.

Answer: We agree that further analysis would be of interest to readers. However, it is not required to state that MAG catalogues include artificial strain diversity. In metagenomic studies, MAGs are considered equivalent to strains in cultivation studies. Therefore, creating MAGs that do not exist directly means that strain-level diversity is inflated. With all the new analyses performed to address the different comments, we feel that this suggestion, whilst interesting, is not mandatory and does not question the relevance of the findings. Due to this comment, we have rephrased the sentence as follows: “These findings suggest that strain-level diversity within MAG catalogues is artificially inflated due to chimerism”. As all datasets generated in this work are made publicly available, we hope it will help others moving forward with their own work and thereby generate even greater added value through additional analyses, as the one suggested above.

- 13- L342-363 The authors should highlight the limitation of their analysis, that is, that they always mapped the reads against the reference genomes of the exact strains they were looking for. This might be similar to some applications of pathogen detection where the exact strain is known, but for general microbiome analyses, usually only the ref. genomes of other strains of the species are available, and genomes of all possible species are included in the database. This might reduce both the sensitivity and also introduce false positives, leading to a lower signal-to-noise ratio in the lower seq depths than the ideal case presented here. It should also be mentioned that k-mer or marker gene-based approaches will have different merits and limitations compared to the full genome mapping considered here.

Answer: As also suggested by Reviewer 3, we have now performed additional analyses by using MetaPhlAn4 to generate “unsupervised” taxonomic profiles (see our other response below). Nevertheless, we have also added a new paragraph about the limitations of our study at the end of the discussion (L511-528), where we included the points mentioned here by the reviewer: “Whilst we used MetaPhlAn4 for some of the analyses, most results were obtained using the exact strains (i.e., their genomes) as a reference, which represents an ideal case for taxonomic and functional readouts”; “We have tested multiple approaches, including standard workflows used by many, to study the chimeric MAGs. We do not exclude that additional strategies not tested in this work may perform better. Future benchmarking of bioinformatic tools, as in Critical Assessment of Metagenome Interpretation (CAMI), will be helpful to move forward with comprehensive testing of new approaches to address chimeric MAGs at scales that go beyond our single study”.

- 14- The authors show that library preparation and presence of background DNA significantly impact the results, especially at lower seq depths. It would be nice to discuss more about the influence of these factors, for example, does lower template DNA or more PCR cycles specifically lead to biases or reduced coverage? The manuscript mentions that Facility 1, using more template DNA and fewer PCR cycles, yielded more robust profiles. It would be valuable to discuss whether differences between facilities could also be due to other factors not taken into account (e.g., bead cleanup?).

Answer: As with the previous point about the impact of wet lab versus bioinformatic analysis, it is not possible to study the individual factors that might have led to differences between the facilities. The main advantage of including several facilities (a third facility was involved during the revision to strengthen the MAG analysis further) was twofold: (1) to demonstrate that differences exist, of which people should be aware when conducting large-scale, multicenter studies, for example; and (2) to confirm the primary findings (the outcome of shallow metagenomics and MAG analysis) in a different setting to bolster confidence. We have considered the comment above in the 'Limitations' section of the manuscript.

Minor points

- 15- The title should include “shallow sequencing” since that is the target application of the benchmark

Answer: Thank you. We have considered this comment and modified the title as follows: “Benchmarking of shotgun sequencing depth highlights limitations of strain-level analysis and shallow metagenomics”

16- L61: "Given the urgent need to better define the strengths and limitations of shallow metagenomics, we systematically evaluated..." This sentence appears twice consecutively.

Answer: Thank you for noticing this mistake, which was corrected.

17- L111: "biobython13" should likely be "Biopython".

Answer: Thank you and sorry; this has been corrected.

18- L185-186: "Thomasclavelia ramosa CLA-JM-H52 and Enterocloster clostridioformis CLA-JM-H51, which were added to the Mock with highest DNA concentration (17.2%; 40 ng), and *V. intestinalis* CLA-AV-13, added at a low concentration (0.86%, 2 ng)." Ensure clarity that *V. intestinalis* reaching 100% coverage despite low input is a notable point, perhaps linked to its genome characteristics or sequencing efficiency.

Answer: This is a good suggestion. However, due to the addition of several datasets, this paragraph had to be rewritten and shortened. While the impact of genome characteristics on sequencing coverage is an interesting topic, it is beyond the scope of this article. Main genome features (size, number of contigs, G+C%) are provided in the Suppl. Table S2, and all genomes are publicly available for further analyses by others.

19- L221: "curves overlay" could be rephrased for clarity, e.g., "their coverage increase curves overlaid".

Answer: Thank you, this was changed as suggested.

20- L277: "KEEG-Decoder" should likely be "KEGG-Decoder". This is repeated on line 281.

Answer: Thank you, this has been corrected.

21- L213-223 This analysis is not yet concerning the de-novo assembly but rather still the mapping-based coverage and thus could be part of the previous paragraph. Similarly, the panels Fig 2 A, B could be part of Fig 1 or removed as they are only a different visualisation of a part of the heatmap in Fig 1A.

Answer: Thank you for the suggestion. Fig 2 A and B have been moved to Suppl. Fig. S7. Due to the interesting findings at the strain-level, we think it is better to keep the data obtained by reference genome mapping and de-novo assembly/binning in the same section. The header of this section has been changed to "Strain-level analysis: *De-novo* MAG reconstruction requires deep sequencing and generates chimeras" to include the mapping-based analysis, which is presented first.

22- Fig 2C the title Mock-log should probably be mock-stag

Answer: Thank you, has been corrected.

23- L284 same log vs. stag

Answer: Thank you, has been corrected.

Reviewer #3 (Remarks to the Author):

Referee #3: Omics methods development

This is an excellent research study done on somewhat complex mock communities, aimed at studying the interplay between sequencing coverage and strain variation on taxonomic, MAG, and functional analysis of microbial communities.

The major results are that taxonomic analysis of these communities *should* be accurate, given the coverage and recovery of the known source genomes in the metagenomic sequencing; that strain-resolved MAG analysis is challenging, at best, and baldly inaccurate, at worst; and that functional analysis is challenging at low coverage.

The conclusions are well supported by the data and the analysis, and overall this is an excellent study that cleanly and substantially expands the literature in this important area.

To the extent that I have any critiques, they are mostly about the presentation of the work in the context of the larger field.

Answer: Thank you for your very positive opinion about our study and for the suggestions. We have considered them (together with all other comments by the other two reviewers) and improved the manuscript accordingly. Please find our responses to your specific comments below.

- 1- First, the strain complexity of these artificial communities should be compared to the expected strain complexity of real communities. My sense is that ~70 strains is reasonably comparable to the expected strain complexity of the dominant species in gut microbiota, but I would like the authors to make a statement here.

Answer: Despite being an easy question, 'How many species colonize the distal gut of one human subject under healthy conditions?', there is no clear answer. Current estimates suggest 300–400 species. In contrast, it is clearer how many species can be detected using sequencing techniques: around 150–200 per individual, depending on the sample, method used and, sequencing depth [Faith *et al.* 2013; Qin *et al.* 2010]. The most complex *in vivo* model community includes approximately 100 members studied *in vitro* and in mice [Cheng *et al.* 2022]. Based on this, the most complex mock community that we used (70 bacterial strains with a staggered distribution) has a lower diversity, but is still covers the majority of functions within a gut microbiomes and is more complex than most examples in the literature. Following Reviewer 2's comments, we have added a new limitations section to the end of the discussion. This point is mentioned there (L511-528).

Faith JJ *et al.* The long-term stability of the human gut microbiota. *Science*. 2013 Jul 5;341(6141):1237439.

Qin J *et al.* A human gut microbial gene catalogue established by metagenomic sequencing. *Nature*. 2010 Mar 4;464(7285):59-65.

Cheng AG *et al.* Design, construction, and *in vivo* augmentation of a complex gut microbiome. *Cell*. 2022 Sep 15;185(19):3617-3636.e19.

- 2- Second, the authors do not use a taxonomic workflow directly, but rather estimate the degree to which the known genomes are sequenced in the mixture. This is not an uninteresting question, but it only gets at one aspect of the situation: is the information there? A different question is: do

taxonomic profilers get distracted by the reference database? Ideally the authors could use a system like Kraken or sourmash (which, in my expert opinion, should be able to recover the strains!) to ask how well a straightforward taxonomy tool actually works. I do recognize that this is in itself a lot of extra work so I would be quite satisfied if the authors simply noted that actual calling of taxonomy on this community might not work that well, given other considerations such as the reference database.

Answer: Thank you for this suggestion. We performed additional analyses and have used MetaPhlAn4 to generate “unsupervised” taxonomic profiles. A comparison of the reference genome-based approach and MetaPhlAn4 is provided in Fig. 1C, and relative abundances per strain in the three Mock communities are shown in Suppl. Fig. S6. Overall, this approach was less sensitive than using the reference genomes and showed higher variance from the targeted relative abundances, especially at lower sequencing depths.

Reviewer #1 (Remarks to the Author):

I appreciate the considerable work that has gone into improving the manuscript, in particular the additional data generated for confirmation and statistical power as well as the additional analyses performed. I also appreciate the thought put into assessing the limitations and offering useful suggestions for microbiome researchers. The new version is much improved but I have a few more questions and suggestions noted below.

Abstract line 29: "...even high-quality MAGs were chimeric, with 54.5 to 81.8% accurately representing the original strains..." Maybe "high-quality" should be in quotes here? Or else say "even MAGs deemed high quality by standard metrics"?

Thanks for the suggestion. The text has been changed accordingly.

Abstract line 32: "Functionally, 2 Gb provided reliable insights at the pathway level..." I still disagree with asserting that a certain amount of sequence is sufficient without noting that this is only known to be true for the communities examined, e.g. "Functionally, 2 Gb provided reliable insights at the pathway level for each of the communities tested..."

Thanks for pointing out this missing detail. The limitation is now expressed by adding "for each of the mock communities tested" to the abstract.

Line 41: "...at an ever-increasing sample size." Do you mean sample number?

Wording was changed to "... at an ever-increasing number of samples."

Line 251: "For Mock-even-70, relative abundances oscillated around the expected value..." I think "oscillated" is a poor choice here, something like "clustered" would be better.

The paragraph was rephrased.

Response to reviews:

On response 10 about the depth of sequencing – it's true that, for example, 333,333 paired-end 150 bp read pairs will generate 0.1 Gb of raw sequence. However, if those paired-end 150 bp reads are generated on 250 bp inserts, they cannot generate more than 250 bp of unique sequence even if 300 high-quality bases are produced. It's not a critical point, but the coverage obtained with 0.1 Gb raw sequence would be different if, for example, you were to sample 666,666 single-ended reads from the dataset.

We see your point. We choose to use a very common sequencing protocol (Illumina PE 150) to make the results broadly applicable. To include the parameter stated, investigations beyond the scope of our work would be needed, which e.g. also include quality loss at the end of the reads for single-reads.

Reviewer #2 (Remarks to the Author):

The revised manuscript has improved significantly, and this reviewer thanks the authors for their effort in updating their analyses.

Reviewer #3 (Remarks to the Author):

I appreciate the substantial revisions and have no further comments. Thank you!